# IRGEN: GENERATIVE MODELING FOR IMAGE RETRIEVAL

## ABSTRACT

While generative modeling has become prevalent across numerous research fields, its potential application to image retrieval has yet to be thoroughly justified. In this paper, we present a novel approach, reframing image retrieval as a variant of generative modeling and employing a sequence-to-sequence model. This provides promising alignment with the overarching theme of unification in current research. Our framework enables end-to-end differentiable search, leading to superior performance through direct optimization. During the development of IRGen, we tackle the key technical challenge of converting an image into a concise sequence of semantic units, which is essential to facilitate efficient and effective search. Extensive experiments demonstrate that our model yields significant improvement over various widely utilized benchmarks, and further validate its performance on million-scale datasets. The most intriguing finding lies in the substantial enhancement of precision scores achieved through generative modeling, which potentially opens the avenue to excluding the rerank stage typically utilized in practical retrieval pipelines.

## 1 INTRODUCTION

Generative modeling has made significant progress in a wide range of tasks including machine translation (Vaswani et al., 2017), conversational modeling (Devlin et al., 2018; Brown et al., 2020; Ouyang et al., 2022; Adiwardana et al., 2020), image captioning (Yu et al., 2022a), image classification (Chen et al., 2020), text-to-image synthesis (Ramesh et al., 2022; Ding et al., 2022), and many more. Originating from language and then expanding to other modalities with specially designed tokenizers, such a universal modeling approach provides a promising direction for unifying different tasks into a versatile pretrained model, which has attracted widespread attention (Alayrac et al., 2022; Wang et al., 2022a; Ouyang et al., 2022; Li et al., 2023). Yet its potential in image retrieval has been unexplored. This paper aims to take the unified trend one step further and investigates generative modeling for image retrieval.

A practical retrieval system generally consists of two stages: feature representation learning (El-Nouby et al., 2021; Liu et al., 2022; Lee et al., 2022b; Yang et al., 2021) and Approximate Nearest Neighbor (ANN) search (Babenko & Lempitsky, 2014b; Johnson et al., 2019; Guo et al., 2020; Chen et al., 2021). Most image retrieval methods focus only on one individual stage while ignoring the fact that both stages are inherently and deeply connected in actual service. Thus, the practical system often requires careful per-task hyperparameter tuning to make the most out of the coordination of the feature extraction and ANN search. While recent progress (Gao et al., 2020; De Cao et al., 2020; Wang et al., 2022b; Tay et al., 2022) have been made towards end-to-end search in the scenario of recommendation, entity retrieval and document retrieval, little has been done for image retrieval.

In this paper, we cast image retrieval as a form of generative modeling and make use of standard Transformer architecture, as in GPT (Brown et al., 2020; Radford et al., 2019; 2018), to enable end-to-end differentiable search. Our model, IRGen, is a sequence-to-sequence model that, given a provided query image, directly generates identifiers corresponding to the query's nearest neighbors. Specifically, the model takes a query image as input and autoregressively predicts discrete visual tokens, which are considered as the identifier of an image. The predicted visual tokens are supposed to point to the query image's nearest neighbor. As such, IRGen can be trained directly from the final retrieval target starting with raw images.

Two fundamental concerns need to be addressed to enable efficient and effective image retrieval using generative modeling. First, autoregressive generative modeling is notable for its slow sampling process due to the inherently sequential nature, thus the run-time cost for retrieval grows at least linearly with respect to the length of a sequence. Second, it is particularly difficult to model the semantic relationship between the identifiers if we drastically shortened the image identifier. Therefore, a semantic tokenizer specially designed for image retrieval is an immediate problem.

We observe that existing image tokenizers (Van Den Oord et al., 2017; Lee et al., 2022a), normally designed for image generation task, are not suitable for image retrieval task, leading to poor performance as analyzed in our experiments. We hence propose several key ingredients that first inject semantic information by applying image-level supervision rather than pixel-level reconstructive supervision, then generate dependent tokens in a sequence by leveraging the recursive property of residual quantization, and lastly ensure fast inference speed by tremendously reducing the length of the sequence via exploiting the global feature instead of spatial patch embeddings. Afterwards, we intentionally adopt the standard Transformer architecture so that it is easy to scale up the model using existing techniques and infrastructures.

The proposed IRGen model has set new records across a diverse range of image retrieval datasets, owing to its end-to-end differentiable search capability. It consistently outperforms previous strong competitors by a significant margin, sometimes even surpassing linear scan search methods. For instance, when compared with the best baseline methods, which include linear scan search, our model achieves remarkable improvement, such as a 20.2% increase in precision@10 on In-shop Clothes dataset (Liu et al., 2016), a 6.0% boost in precision@2 on CUB200 (Wah et al., 2011) and a 2.4% enhancement in precision@2 on Cars196 (Krause et al., 2013). To assess the scalability of our model, we further experiment on million-level datasets, namely ImageNet (Deng et al., 2009) and Places365 (Zhou et al., 2017a), and consistently demonstrated superior performance in these challenging scenarios.

It is our belief that generative models have the potential to redefine the landscape of image retrieval. The application of generative modeling to image retrieval tasks not only represents an exciting opportunity but also has the potential to unify information retrieval across various modalities. At a technical level, our model, IRGen, effortlessly bridges the gap between feature representation learning and approximate search, creating an end-to-end differentiable framework that enables direct optimization based on retrieval objectives. Furthermore, the entire framework is conceptually straightforward, with all components relying on the standard Transformer architecture, renowned for its remarkable scalability (Du et al., 2022; Chowdhery et al., 2022; Shoeybi et al., 2019; Xu et al., 2021). To the best of our knowledge, our work marks the pioneering exploration of generative modeling in the domain of image retrieval, extending the boundaries of generative modeling into new territories. Along this journey, we have introduced a fundamentally distinct retrieval approach that has demonstrated impressive performance on various retrieval benchmarks.

## 2 METHOD

### 2.1 SEMANTIC IMAGE TOKENIZER

As Transformer becomes the ubiquitous architecture in computer vision, it has emerged many successful image tokenizers such as VQ-VAE (Van Den Oord et al., 2017; Ramesh et al., 2021; Gafni et al., 2022; Yu et al., 2021), RQ-VAE (Lee et al., 2022a) and so on. Basically, these methods learn a variational auto-encoder with discrete latent variables, together with a learnable and indexable codebook over a collection of raw images. As a result, an image is represented as a sequence of accountable discrete codes indicating the entries in the codebook. A proper combination of entries can be decoded to a high-quality image through the decoder. Such tokenizer has been widely applied to image synthesis, and can be easily extended to audio and video synthesis.

Despite its success in image generation, we believe that this approach may not be well-suited for the retrieval task. There are several reasons for this. Firstly, the process of decoding latent codes to reconstruct raw images is essential for generating images in synthesis tasks but is not required for retrieval tasks. Secondly, the length of the code sequence has a significant impact on the inference speed of autoregressive models, which is crucial for efficient search in our case. It is particularly challenging to handle very short sequences of codes, whereas current code sequences used for re-

trieval are often quite long (e.g., the feature map of $8 \times 8$ with a depth of 4 in RQ-VAE results in a sequence length of 256). Additionally, for retrieval, it's important to inject semantic information into the latent codes, while image reconstruction loss, which is commonly used in generative models, tends to focus on low-level details, including imperceptible local details and noise.

Building on the insights mentioned earlier, we suggest investigating the global feature extracted from the class token instead of relying on the default spatial tokens. This approach offers a substantial reduction in sequence length (from 64 tokens down to just 1 token). Additionally, the class token inherently contains a condensed, high-level semantic representation as a byproduct of this strategy. Let $\mathbf{f}_{cls}$ denote the $d$-dimensional feature vector outputted from the class token, which is taken as the image representation. We adopt residual quantization (RQ) or stacked composite quantization to approximate this feature. Suppose there are $M$ codebooks with each containing $L$ elements, $\mathcal{C}_m = \{\mathbf{c}_{m1}, \cdots, \mathbf{c}_{mL}\}$, RQ recursively maps the embedding $\mathbf{f}_{cls}$ to a sequentially ordered $M$ codes, $\mathbf{f}_{cls} \rightarrow \{l_1, l_2, \cdots, l_M\} \in [\mathbb{L}]^M$. Let $\mathbf{r}_0 = \mathbf{f}_{cls}$, we have

$$l_m = \arg\min_{l \in [\mathbb{L}]} \|\mathbf{r}_{m-1} - \mathbf{c}_{ml}\|_2^2, \tag{1}$$

$$\mathbf{r}_m = \mathbf{r}_{m-1} - \mathbf{c}_{ml_m}, \ m = 1, 2, \cdots, M. \tag{2}$$

The process of sequentially generating discrete codes is inherently compatible with sequential autoregressive generation. This alignment helps alleviate the optimization challenges associated with modeling the relationships within identifiers.

To further inject semantic prior, we train the network under classification loss over both the original embeddings as well as the reconstructed embeddings. In particular, we consider a series of reconstruction levels denoted as $\hat{\mathbf{f}}_{cls}^{\leq m} = \sum_{i=1}^{m} \mathbf{c}_{il_i}, m = 1, 2, \cdots, M$. Each prefix code thus encodes semantic information to a certain degree. Adding up the $M$ levels of partial reconstruction error, the complete objective function is then formulated as,

$$\mathcal{L} = \mathcal{L}_{cls}(\mathbf{f}_{cls}) + \lambda_1 \sum_{m=1}^{M} \mathcal{L}_{cls}(\hat{\mathbf{f}}_{cls}^{\leq m}) + \lambda_2 \sum_{m=1}^{M} \|\mathbf{r}_m\|_2^2, \tag{3}$$

$$\mathbf{r}_m = \mathbf{f}_{cls} - \mathrm{sg}[\hat{\mathbf{f}}_{cls}^{\leq m}], \ m = 1, 2, \cdots, M, \tag{4}$$

where $\mathrm{sg}[\cdot]$ is the stop gradient operator. During training, we adopt alternative optimization to update the codebook and the network. For computing the gradient of $\mathcal{L}_{cls}(\hat{\mathbf{f}}_{cls}^{\leq m})$, we follow the straight-through estimator (Bengio et al., 2013) as in (Van Den Oord et al., 2017) and approximate the gradient by copying the gradients at $\hat{\mathbf{f}}_{cls}^{\leq m}$ directly to $\mathbf{f}_{cls}$. After optimization, we hope that images with similar classes have close codes. In the experiments, we present comparison with other discrete identifiers including random codes and codes from hierarchical k-means algorithm or from RQ-VAE.

## 2.2 ENCODER-DECODER FOR AUTOREGRESSIVE RETRIEVAL

Once we have established a robust discrete latent structure equipped with semantic prior information, our next step is to train a powerful autoregressive sequence-to-sequence model solely on these discrete random variables without referring their visual content. Our encode-decoder architecture decouples the input embedding from the generation of discrete codes. The model begins by taking a query image as input to obtain the query embedding, which is then used to produce the discrete codes. It is worth noting that the yielded discrete codes represent the query's nearest neighbor images within the database. Therefore, our training process involves image pairs $(x_1, x_2)$, where $x_2$ is the nearest neighbor of $x_1$. Our model's objective is to predict the identifiers of $x_2$ when given $x_1$ as input. This setup allows the model to learn the semantic relationships between images in the dataset. Figure 1 provides a concise view of our training pipeline.

To be specific, let the encoder be denoted as $\mathbb{E}$ based on the ViT base architecture and the decoder be $\mathbb{D}$, a standard Transformer that includes causal self-attention, cross-attention and MLP layers. We leverage the spatial tokens outputted from the encoder as the input embedding, $\mathbf{e} = \mathbb{E}(x_1)$, which is injected into the decoder through cross attention. Our training objective involves predicting the next token in the image identifier sequence. Specifically, we aim to maximize the probability of the $i$-th token of the image identifier given the input embedding and the previously predicted tokens, $p(l_i|x_1, l_1, \cdots, l_{i-1}, \theta)$, where $\theta$ denotes the parameters of both $\mathbb{D}$ and $\mathbb{E}$, and $l_1, l_2, \cdots, l_M$ are the $M$ tokens that make up the image identifier for $x_2$, generated by the image tokenizer. By

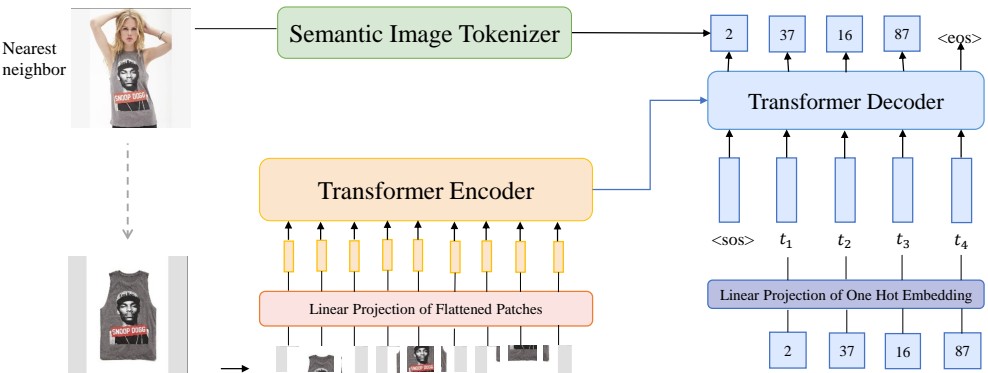

Figure 1: We train a sequence-to-sequence model to autoregressively predict the nearest neighbor's image identifiers given a query image..

maximizing the probability of each token, we effectively maximize the likelihood of generating the image identifier of an image,

$$p(l_1, \cdots, l_M | x_1, \theta) = \Pi_{m=1}^{M} p(l_i | x_1, l_1, \cdots, l_{m-1}, \theta). \tag{5}$$

We apply a softmax cross-entropy loss on a vocabulary of $M$ discrete tokens during training. This loss guides the model to generate the correct sequence of tokens that represent the image identifier.

## 2.3 BEAM SEARCH

During inference, given a query image $\mathbf{q}$, we first calculate the query embedding processed by the encoder $\mathbb{E}$ and then generate the discrete codes through the decoder $\mathbb{D}$ based on the query embedding in an autoregressive manner. These discrete codes represent an image that is considered as the nearest neighbor to the query image. To perform Top-K retrieval, our model can use beam search, allowing us to find the top-$K$ images that are the closest matches to the query image.

Specifically, when provided with a query image, our model employs an autoregressive approach to generate discrete codes, commencing with the start-of-sequence token. In contrast to the single candidate considered in greedy search, our model utilizes beam search, which maintains a "beam" of the top-K candidates at each generation step. At each step, the candidates are ranked based on their scores, which are computed as the product of the probabilities associated with their individual elements. We retain only the top-K sequences with the highest scores.

It's important to note that not all generated identifiers are necessarily valid, meaning that an identifier belonging to the set $[\mathbb{L}]^M$ may not correspond to any images within the retrieval database. Therefore, we must validate the generated image identifier at each step, which can be a time-consuming process. However, we address this challenge by expediting the validation process. We achieve this by imposing constraints on the beam search, ensuring that it explores only within a prefix tree containing valid codes. This optimization enhances the efficiency of the retrieval process.

**Beam search vs. ANN search.** Indeed, there are certain similarities between beam search and approximate nearest neighbor (ANN) search, as both methods aim to select the top-K most promising candidates by traversing tree-like data structures. However, they diverge significantly in how they calculate the score to choose the current node. In ANN search, the score is typically determined by computing the distance between the query feature and the feature associated with the node, using a specific distance metric. On the other hand, in beam search, the score or probability is generated as a function through a differentiable neural network, often an autoregressive model. This neural network is conditioned on the query and learns to estimate the score or probability of a candidate sequence. Consequently, the entire retrieval pipeline can be optimized in an end-to-end manner, taking advantage of neural network training techniques.

| Model | In-shop | | | | CUB200 | | | | Cars196 | | | |
|---|---|---|---|---|---|---|---|---|---|---|---|---|
| | 1 | 10 | 20 | 30 | 1 | 2 | 4 | 8 | 1 | 2 | 4 | 8 |
| *Linear scan search* | | | | | | | | | | | | |
| Res101-Img | 30.7 | 10.2 | 7.1 | 5.8 | 46.8 | 43.6 | 39.9 | 34.9 | 25.9 | 22.0 | 18.5 | 15.4 |
| CLIP | 57.5 | 22.8 | 16.6 | 14.1 | 66.0 | 63.5 | 59.4 | 53.8 | 70.8 | 67.8 | 63.3 | 57.2 |
| CGD$_{(repro)}$ | 83.2 | 47.8 | 40.2 | 37.0 | 76.7 | 75.5 | 73.7 | 71.4 | 87.1 | 86.1 | 84.6 | 82.6 |
| IRT$_{R(repro)}$ | 92.7 | 59.6 | 51.1 | 47.6 | 79.3 | 77.7 | 75.0 | 71.4 | 75.6 | 73.1 | 68.3 | 61.7 |
| FT-CLIP | 91.4 | 66.8 | 58.9 | 55.4 | 79.2 | 77.6 | 76.0 | 73.2 | 88.4 | 87.7 | 87.1 | 85.8 |
| *Faiss IVF PQ search* | | | | | | | | | | | | |
| CGD$_{(repro)}$ | 60.4 | 30.5 | 24.5 | 22.0 | 71.6 | 70.8 | 69.9 | 68.7 | 84.8 | 84.4 | 84.1 | 83.3 |
| IRT$_{R(repro)}$ | 68.6 | 35.7 | 29.3 | 26.6 | 68.9 | 67.6 | 66.2 | 63.4 | 59.1 | 57.5 | 54.7 | 51.7 |
| FT-CLIP | 63.7 | 37.0 | 30.7 | 28.0 | 72.6 | 72.1 | 71.2 | 69.7 | 86.5 | 86.3 | 86.2 | 86.0 |
| *ScaNN search* | | | | | | | | | | | | |
| CGD$_{(repro)}$ | 83.0 | 47.7 | 40.3 | 37.2 | 76.7 | 75.2 | 73.8 | 71.4 | 87.1 | 86.1 | 84.5 | 82.6 |
| IRT$_{R(repro)}$ | 92.0 | 58.2 | 50.0 | 46.6 | 79.3 | 77.7 | 75.1 | 71.4 | 75.4 | 72.8 | 68.1 | 61.6 |
| FT-CLIP | 90.4 | 64.6 | 56.9 | 53.5 | 79.2 | 77.5 | 76.0 | 73.2 | 88.3 | 87.7 | 87.1 | 85.8 |
| *SPANN search* | | | | | | | | | | | | |
| CGD$_{(repro)}$ | 83.0 | 47.7 | 40.3 | 37.1 | 76.7 | 75.5 | 73.7 | 71.4 | 87.0 | 86.1 | 84.6 | 82.6 |
| IRT$_{R(repro)}$ | 91.4 | 56.2 | 47.9 | 44.5 | 79.3 | 77.6 | 75.0 | 71.4 | 74.8 | 72.4 | 67.6 | 61.1 |
| FT-CLIP | 90.2 | 62.9 | 55.1 | 51.8 | 78.5 | 77.6 | 76.0 | 73.2 | 88.6 | 88.1 | 87.5 | 86.3 |
| *Beam search* | | | | | | | | | | | | |
| IRGen (ours) | **92.4** | **87.0** | **86.6** | **86.5** | **82.7** | **82.7** | **83.0** | **82.8** | **90.1** | **89.9** | **90.2** | **90.5** |

Table 1: Precision comparison with different baselines, for which we consider linear scan search, Faiss IVF search and SPANN search. (repro) denotes the model reproduced by ourselves to ensure the same data process and comparable model size for fair comparison. Our model adopt beam search for retrieval, achieving significant improvement and performing even better than linear scan search.

## 3 EXPERIMENTS

We conduct comprehensive evaluations to demonstrate the performance of the proposed IRGen. We first evaluate our method on common image retrieval datasets and on two large-scale datasets, ImageNet (Deng et al., 2009) and Places365 (Zhou et al., 2017a). For a detailed description of the datasets and implementation details, please refer to the appendix.

**Baselines.** We evaluate our model's performance in comparison to five competitive baselines: 1) ResNet-101 (He et al., 2016) trained from ImageNet dataset, denoted as Res101-Img, which is commonly used as a feature extraction tool for various tasks; 2) CLIP (Radford et al., 2021) trained on 400M image-text pairs, known for powerful zero-shot capability; 3) CGD (Jun et al., 2019), a state-of-the-art method based on ResNet; 4) IRT (El-Nouby et al., 2021), a Transformer-based model for image retrieval and we use the best-performing model IRT$_R$; 5) FT-CLIP, a baseline finetuned from CLIP on the target dataset. For both CGD and IRT, we have reproduced these models to ensure consistent data processing and comparable model sizes. Specifically, we use ResNet-101 for CGD and DeiT-B for IRT. We also provide their best results from their original papers for reference.

**Search process.** The baseline models primarily focus on effective feature learning. After training, these models are utilized to extract features for the database images. During the search process, a given query image is initially passed through the model to obtain its query feature. Subsequently, this query feature is compared with the features of database images using a specific distance metric. Following the conventions established in previous works such as (Radford et al., 2021; Jun et al., 2019; El-Nouby et al., 2021), we employ the cosine distance for the CLIP model and the Euclidean distance for the other baseline models. We evaluate two search strategies: linear scan search (K-nearest neighbors or KNN) and approximate nearest neighbor search (ANN). Linear scan search is known for its accuracy but is computationally intensive. In contrast, ANN is significantly more efficient. For ANN, we explore: (i) the popular Faiss IVF PQ (Johnson et al., 2019); (ii) the state-of-the-art memory-based algorithm ScaNN (Guo et al., 2020) with the default setting; and (iii) the state-of-the-art disk-based SPANN algorithm (Chen et al., 2021). These evaluation strategies allow us to assess the retrieval performance of our model against a variety of search methods.

### 3.1 RESULTS

Table 1 presents a detailed performance comparison in terms of precision@$K$, which assesses the percentage of retrieved candidates that share the same class as the query among the top $K$ results.

| Model | In-shop | | | | CUB200 | | | | Cars196 | | | |
|---|---|---|---|---|---|---|---|---|---|---|---|---|
| | 1 | 10 | 20 | 30 | 1 | 2 | 4 | 8 | 1 | 2 | 4 | 8 |
| *Linear scan search* | | | | | | | | | | | | |
| Res101-Img | 30.7 | 55.9 | 62.7 | 66.8 | 46.8 | 59.9 | 71.7 | 80.8 | 25.9 | 35.6 | 47 | 59.7 |
| CLIP | 57.5 | 83.0 | 87.5 | 89.7 | 66.0 | 78.1 | 87.7 | 93.5 | 70.8 | 82.6 | 91.1 | 95.9 |
| CGD* | 91.9 | 98.1 | 98.7 | 99.0 | 79.2 | 86.6 | 92.0 | 95.1 | 94.8 | 97.1 | 98.2 | 98.8 |
| $IRT_R$* | 91.9 | 98.1 | 98.7 | 99.0 | 76.6 | 85.0 | 91.1 | 94.3 | - | - | - | - |
| FT-CLIP | 91.4 | 97.3 | 98.1 | 98.5 | 79.2 | 85.0 | 89.3 | 92.0 | 88.4 | 90.5 | 92.5 | 93.8 |
| *Faiss IVF PQ search* | | | | | | | | | | | | |
| $CGD_{(repro)}$ | 60.4 | 76.0 | 77.1 | 77.4 | 71.6 | 77.4 | 81.5 | 84.2 | 84.8 | 88.0 | 89.8 | 91.0 |
| $IRT_{R(repro)}$ | 68.6 | 79.2 | 80.0 | 80.2 | 68.9 | 77.9 | 85.0 | 89.3 | 59.1 | 70.4 | 78.2 | 83.4 |
| FT-CLIP | 63.7 | 70.7 | 71.1 | 71.2 | 72.6 | 78.0 | 82.3 | 85.2 | 86.5 | 86.9 | 87.2 | 87.5 |
| *ScaNN search* | | | | | | | | | | | | |
| $CGD_{(repro)}$ | 83.0 | 94.8 | 96.2 | 96.7 | 76.7 | 83.5 | 88.0 | 91.8 | 87.1 | 91.7 | 94.6 | 96.6 |
| $IRT_{R(repro)}$ | 92.0 | **97.8** | **98.3** | **98.4** | 79.3 | 86.8 | 91.9 | 94.7 | 75.4 | 84.7 | 90.9 | 95.0 |
| FT-CLIP | 90.4 | 95.9 | 96.6 | 96.9 | 79.2 | 85.0 | 89.2 | 92.7 | 88.3 | 90.5 | 92.4 | 93.7 |
| *SPANN search* | | | | | | | | | | | | |
| $CGD_{(repro)}$ | 83.0 | 95.0 | 96.4 | 96.9 | 76.7 | 83.4 | 87.9 | 91.8 | 87.0 | 91.7 | **94.6** | **96.7** |
| $IRT_{R(repro)}$ | 91.4 | 97.2 | 97.6 | 97.7 | 79.3 | **86.8** | **91.9** | **94.7** | 74.8 | 84.3 | 90.5 | 94.7 |
| FT-CLIP | 90.2 | 95.8 | 96.7 | 97.0 | 78.5 | 85.0 | 89.4 | 92.9 | 88.6 | 90.7 | 92.5 | 94.2 |
| *Beam search* | | | | | | | | | | | | |
| IRGen (ours) | **92.4** | 96.8 | 97.6 | 97.9 | **82.7** | 86.4 | 89.2 | 91.4 | **90.1** | **92.1** | 93.2 | 93.7 |

Table 2: Recall comparison with different baselines, for which we consider linear scan search, Faiss IVF search and SPANN search. (repro) denotes the model reproduced by ourselves to ensure the same data process and comparable model size for fair comparison. we include the best result of CGD and IRT from their original papers for context with * denotation. Our model adopt beam search for retrieval, achieving comparable performance in most cases.

Our model consistently outperforms all other models, even surpassing those employing linear scan search. Notably, we achieve remarkable improvements, such as a 20.2% boost in precision@10 on the In-shop Clothes dataset, a 6.0% increase in precision@2 on the CUB200 dataset, and a 2.4% gain in precision@2 on the Cars196 dataset. Furthermore, several observations can be made: 1) Finetuned models, tailored to specific datasets, exhibit significantly better performance compared to off-the-shelf feature extractors like CLIP and ImageNet-pretrained ResNet-101. 2) Generally, models equipped with ANN algorithms perform slightly worse than their counterparts using linear scan search. However, there are exceptions, such as FT-CLIP with SPANN search on the Cars196 dataset, which demonstrates the importance of end-to-end optimization. 3) Our model consistently maintains high precision scores as $K$ increases, while other models experience a substantial drop.

Table 2 provides a comparison of different models using the Recall@$K$ metric. Recall@$K$ measures the proportion of queries for which at least one image among the top $K$ retrieved candidates shares the same label as the query image, yielding a score of 1 if true and 0 otherwise. The table also includes the best recall results of CGD and IRT from their respective original papers for reference. It's important to note that these models may have different data preprocessing, model sizes, and additional training techniques. Here are the key observations: 1) Our IRGen model achieves the highest Recall@1 score compared to all other models. However, for other recall scores, our model performs similarly or slightly worse. This discrepancy may arise from the current objective loss used in autoregressive models, which heavily optimizes for Recall@1 while giving less emphasis to other recall values. One potential solution is to incorporate the beam search process into training for joint optimization. 2) Different combinations of feature extractors and ANN algorithms exhibit significant variations across the three datasets, highlighting the challenges of achieving coordination in practical scenarios. 3) Notably, despite the high recall achieved by baselines, they often require an additional re-ranking stage to improve precision, whereas our model already attains high precision scores without the need for re-ranking.

Figure 2 illustrates the precision-recall curve, where recall represents the true positive rate. Our approach, IRGen, consistently delivers outstanding performance, maintaining high precision and recall simultaneously. In addition to precision-recall analysis, we evaluate our model using the mean reciprocal rank (MRR) metric, which measures the inverse of the rank of the first relevant item. We compute MRR for four different values: 1, 2, 4, and 8, and display the corresponding curves in Figure 3. The baselines employ the SPANN retrieval algorithm. Our IRGen model consistently outperforms the baselines across all evaluated metrics, confirming the effectiveness of our framework.

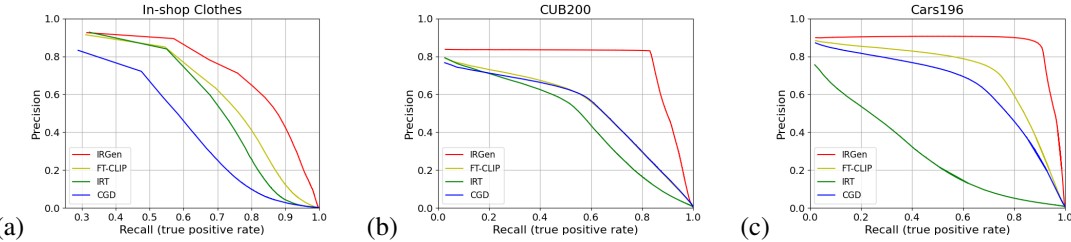

Figure 2: Precision-Recall (TPR) curve comparison for different methods on (a) In-shop Clothes, (b) CUB200 and (c) Cars196 dataset.

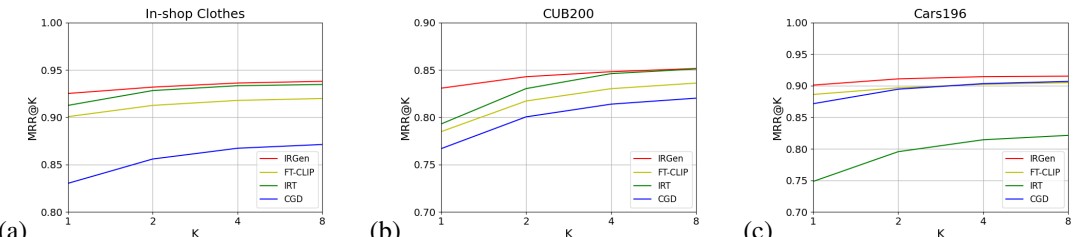

Figure 3: MRR with respect to 1,2,4,8 comparison for different methods on (a) In-shop Clothes, (b) CUB200 and (c) Cars196 dataset.

Notably, there is significant variability in the performance gap between each baseline and our model across the three datasets, highlighting the challenges and dataset-dependent nature of retrieval tasks.

**Results on million-level datasets.** We further experiment our approach with ImageNet dataset (Deng et al., 2009) that contains 1,281,167 images and Places365-Standard (Zhou et al., 2017a) containing about $1.8M$ images from 365 scene categories. We compare with the strong baselines including CLIP model as well as FT-CLIP model finetuned based on CLIP model. The comparison is reported in Figure 4 and Table 3, focusing on precision@$K$ and MAP@100. Our IRGen model consistently outperforms the baselines, achieving the best results in terms of precision@$K$ and MAP@100. The precision values for our model remain consistently high as $K$ increases, while the baselines experience noticeable performance degradation. These results confirm the effectiveness of our model in handling large-scale datasets like ImageNet, where it maintains high precision across varying values of $K$ and outperforms the baseline models.

## 3.2 ABLATIONS

**The effect of identifiers.** In our study of image identifiers, we compared three different approaches: (1) assigning random identifiers to images, (2) hierarchical k-means (HKM), and (3) using the image tokenizer RQ-VAE (Bevilacqua et al., 2022). The results of this comparison are summarized in Table 5. The random assignment of identifiers to images yielded expectedly lower performance. This performance gap can be attributed to the fact that models with random identifiers need to learn not only the interaction between queries and image identifiers but also allocate capacity to learn relationships within the identifiers themselves. On the other hand, HKM showed superior performance compared to random assignment, underscoring the significance of semantic identifiers. However, our proposed semantic image identifiers demonstrated a clear improvement over HKM, highlighting their effectiveness in enhancing retrieval performance. In contrast, the performance of RQ-VAE significantly trailed behind our model, with a performance less than 10 percent. We attribute this difference to that the sequence length in RQ-VAE is too long for the model to effectively capture relationships within the identifiers.

**Generalize to new data.** Addressing the inclusion of fresh data holds particular significance, especially in the context of search scenarios. To assess this capacity, we conducted an experiment where

| Dataset | Model | | |
|---|---|---|---|
| | CLIP | FT-CLIP | IRGen (ours) |
| ImageNet | 44.1 | 65.5 | **76.0** |
| Places365 | 22.1 | 30.3 | **44.3** |

Table 3: MAP@100 comparison on two million-level datasets. The results of CLIP and FT-CLIP are retrieved by SPANN.

| Model | Precision | | |
|---|---|---|---|
| | 1 | 10 | 100 |
| FT-CLIP + Linear Scan | 70.6 | 65.0 | 55.6 |
| IRGen (Ours) | **77.0** | **77.9** | **77.4** |

Table 4: Generalize to new data. We split 5% of the training data from the ImageNet dataset for inference and remained unseen during training.

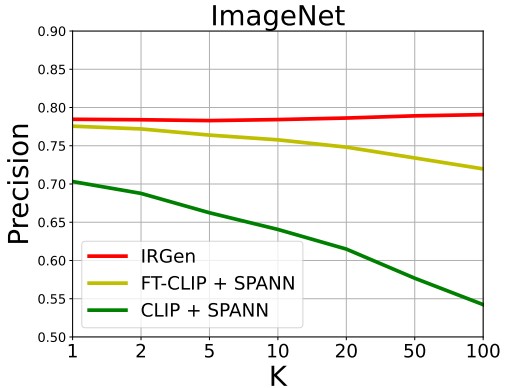
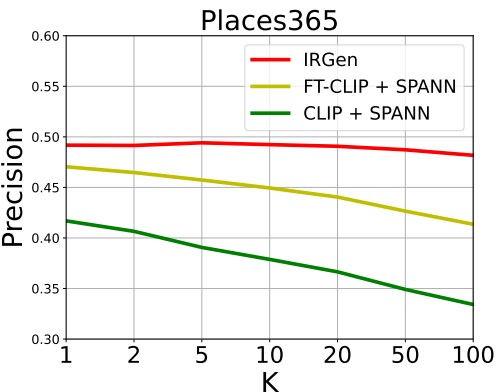

Figure 4: Precision comparison on large scale datasets: ImageNet and Places365.

| Identifier | T | Precision | | | Recall | | |
|---|---|---|---|---|---|---|---|
| | | 1 | 10 | 20 | 1 | 10 | 20 |
| Random | 4 | 87.6 | 75.4 | 70.8 | 87.6 | 95.1 | 96.0 |
| HKM$_{100}$ | 4 | 88.2 | 80.0 | 78.2 | 87.2 | 93.1 | 94.3 |
| HKM$_{200}$ | 3 | 89.0 | 81.6 | 79.8 | 89.0 | 93.9 | 94.9 |
| Ours | 4 | **92.4** | **87.0** | **86.6** | **92.4** | **96.8** | **97.6** |

Table 5: Ablation study on the image identifier (T=length).

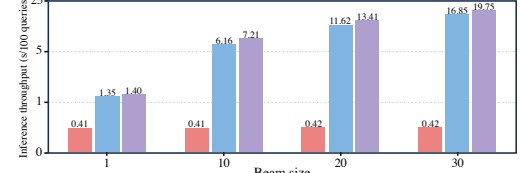

Figure 5: Illustrating the search speed using beam search.

we intentionally withheld 5% of the training data from the ImageNet dataset during the training phase and introduced it during inference, all without updating the existing codebook and autoregressive (AR) model. In this experiment, we compared the performance of our model with the formidable baseline FT-CLIP, which is equipped with a linear scan search. The results, as displayed in Table 4, reveal that our model maintains superior performance even when confronted with new data. This observation highlights our model's remarkable ability to effectively generalize to previously unseen data. This capability is attributed to the fact that our model can derive semantic identifiers for the newly introduced images using the codebook, leveraging the knowledge it has acquired through the autoregressive decoder. The AR model excels in capturing the semantic structure embedded within the database through its learned parameters.

**Inference throughput.** In addition to search accuracy, search efficiency is a critical criterion for retrieval systems. To assess the time cost of our autoregressive (AR) model, we conducted our analysis on an NVIDIA V100-16G GPU. In Figure 5, we present the throughput for 100 queries, with beam sizes set at 1, 10, 20, and 30 for comparison. Additionally, we break down the time cost of each component during retrieval. The results show that the encoder is quite fast, while the autoregressive decoder becomes the major bottleneck, especially as the beam size increases. Additional time is consumed for checking the validity of predictions, as it's possible that the predicted identifier may not exist in the database. Overall, the time cost is within an acceptable range. For instance, it takes approximately 0.07 seconds (with a beam size of 10) or 0.19 seconds (with a beam size of 30) per query. It's important to highlight that our model operates as an end-to-end retrieval method, which doesn't include the re-ranking step. In practical applications, re-ranking is typically necessary to achieve higher precision, but it can significantly increase the time required for the retrieval process.

## 4 RELATED WORK

**Image retrieval.** Traditionally, hand-crafted features are heuristically designed to describe the image content based on its color (Wengert et al., 2011; Wang & Hua, 2011), texture (Park et al., 2002;

Wang et al., 2014b) or shape (Cao et al., 2011). Typical features include GIST (Siagian & Itti, 2007), SIFT (Lowe, 1999), SURF (Bay et al., 2006), VLAD (Jégou et al., 2010) and so on. Recent years have witnessed the explosive research on deep learning based features trained over labeled images. Besides the evolvement of the network architecture designs (Krizhevsky et al., 2017; He et al., 2016; Vaswani et al., 2017), numerous efforts (Wieczorek et al., 2020; El-Nouby et al., 2021) have been dedicated to various loss functions including classification loss (Zhai & Wu, 2018; Zhou et al., 2019), triplet loss (Yuan et al., 2020), contrastive loss (Jun et al., 2019; El-Nouby et al., 2021), center loss (Wieczorek et al., 2020) and so on. The similarity between features can be calculated through some distance measure or evaluated through re-ranking techniques (Revaud et al., 2019).

Another different line of research centers on approximate nearest neighbor search to speed up the search process, accepting a certain level of compromise in search accuracy. One way is to enable fast distance computation through hashing and quantization techniques such as LSH (Indyk & Motwani, 1998), min-Hash (Chum et al., 2008), ITQ (Gong et al., 2012), PQ (Jegou et al., 2010), and many others (Ge et al., 2013; Wang & Zhang, 2018; Zhu et al., 2016). The other way is to reduce the number of distance comparison by retrieving a small number of candidates. Typical methods include partition-based indexing (Babenko & Lempitsky, 2014b; Xia et al., 2013) that partitions the feature space into some non-overlapping clusters and graph-based indexing (Jayaram Subramanya et al., 2019) that builds a neighborhood graph with edges connecting similar images. To improve the recall rate while ensuring fast search speed, hierarchical course-to-fine strategy (Malkov & Yashunin, 2018) has been the popular choice that the retrieved candidates are refined level by level. Additionally, a number of excellent works have introduced hybrid indexing (Chen et al., 2021) that improves search by leveraging the best of both indexing schemes while avoiding their limitations.

**Generative modeling.** Deep autoregressive networks are generative sequential models that assume a product rule for factoring the joint likelihood and model each conditional distribution through a neural network. AR models have shown extremely powerful progress in generative tasks across multiple domains such as images (Chen et al., 2020; Yu et al., 2022b), texts (Radford et al., 2019; Yang et al., 2019), audio (Dhariwal et al., 2020; Chung et al., 2019), and video (Wu et al., 2022; Weissenborn et al., 2019). The particular key component involves linearizing data into a sequence of symbols with notable works such as VQ-VAE (Van Den Oord et al., 2017), RQ-VAE (Lee et al., 2022a). Recently, a number of works (Tay et al., 2022; Wang et al., 2022b; De Cao et al., 2020) further explored the idea of using AR model to empower entity retrieval and document retrieval.

Most related to our work are NCI (Wang et al., 2022b) and DSI (Tay et al., 2022), which are concerned with document retrieval. However, these approaches utilize hierarchical k-means clustering applied to document embeddings derived from a small pretrained language model to obtain document identifiers. In contrast, we put forward a novel approach that involves learning the identifier directly from semantic supervision, and we showcase its effectiveness in the context of image retrieval. We posit that this discovery can also be advantageous for document retrieval tasks.

## 5 CONCLUSION

In this paper, we delve into the realm of generative modeling to enhance end-to-end image retrieval, a process that directly connects a query image to its closest match. With the introduction of our semantic image tokenizer, we've demonstrated that our model excels at achieving remarkable precision without compromising recall. Through extensive ablation studies and evaluations on large-scale datasets, we've underscored the superior performance of our approach. We believe that this innovative approach to generative modeling in image retrieval not only pushes the boundaries of this field but also holds potential for broader applications.

**Limitations.** While our model has shown significant performance improvements, it's important to acknowledge its limitations, which can serve as avenues for future research. Although our model has demonstrated scalability to million-scale datasets, dealing with billion-scale datasets is a complex challenge. It may necessitate even larger models with higher capacity, potentially impacting inference speed. Striking a balance between model capacity and speed is an area that warrants exploration for efficient and effective billion-scale search. Training large autoregressive models requires substantial computational resources, which raises environmental concerns. Research efforts to enable efficient training, such as fast fine-tuning of pretrained models, are crucial to mitigate energy consumption and environmental impact.

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

## A    Experiment Setting

**In-shop Clothes** retrieval dataset (Liu et al., 2016) is a large subset of DeepFashion with large pose and scale variations. This dataset consists of a training set containing 25,882 images with 3997 classes, a gallery set containing 12,612 images with 3985 classes and a query set containing 14,218 images with 3985 classes. The goal is to retrieve the same clothes from the gallery set given a fashion image from the query set. We use both the training set and the gallery set for training in our experiments.

**CUB200** (Wah et al., 2011) is a fine-grained dataset containing 11,788 images with 200 classes belong to birds. There are 5,994 images for training and 5,794 images for testing.

**Cars196** (Krause et al., 2013) is also a fine-grained dataset about cars. It contains 16,185 images with 196 car classes, which is split into 8,144 images for training and 8,041 images for testing.

**ImageNet** dataset (Deng et al., 2009) contains 1,281,167 images for training and 50,000 validation images for testing, in which we randomly sample 5,000 images as queries to speed up the evaluation process.

**Places365-Standard** (Zhou et al., 2017a) includes about $1.8M$ images from 365 scene categories, where there are at most 5000 images per category.

**Implementation details.** We adopt ViT-B for encoder and similar architecture for decoder (12 transformer decoder block with dimension 768). The input image is of resolution $224 \times 224$ and is partitioned to $14 \times 14$ patches with each patch sized $16 \times 16$. Intuitively, a warm initialization of encoder should largely stable the training process. We thus warm-start the model with encoder initialized by the pretrained CLIP model (Radford et al., 2021). We randomly initialize the remaining fully connected layer and the decoder. The semantic image tokenizer is trained with a batch size of 128 on 8 V100 GPUs with 32G memory per card for 200 epochs. We adopt an AdamW optimizer (Loshchilov & Hutter, 2017) with betas as $(0.9, 0.96)$ and weight decay as $0.05$. We use cosine learning rate scheduling. Note that we set the initial learning rate as $5e-4$ for the FC layers. The learning rate of the encoder is set as one percentage of the learning rate of FC layers. We train our models with 20 warming-up epochs and the initial learning rate is $5e-7$. For training autoregressive model, we select similar image pairs $(x_1, x_2)$. Since current retrieval datasets are usually labeled with class information, we randomly sample an image $x_2$ which shares the same class with $x_1$ as the nearest neighbor. For autoregressive model, we use batch size of 64 on 8 V100 GPUs with 32G memory per card for 200 epochs. The optimizer and the scheduler are same as the semantic image tokenizer mentioned above. The initial learning rate is $4e-5$ for the decoder and the learning rate for encoder is always one percentage of that for decoder. The hyperparameter for quantization is set to $M = 4$ and $L = 256$ for fast inference. For ImageNet and Places365, the experimental settings are the same as before except that we enlarge the layer of decoder to 24 to increase the capacity for AR modeling.

## B    Ablation about Sequence Length

We further investigate the length of identifier in our image tokenizer. We experiment different lengths and report the results in Table 6. We can see that if the length of the identifier is too small (for example 2), the model gets inferior performance. As with the length gets longer to 4 or 6, the model gets better performance. At last the performance drops a little bit if the length is too long (8). We think 4-6 would be a good choice in most cases and we simply use 4 in all our experiments.

| T | Precision | | | | Recall | | | |
|---|---|---|---|---|---|---|---|---|
| | 1 | 10 | 20 | 30 | 1 | 10 | 20 | 30 |
| 2 | 72.1 | 69.6 | 68.9 | 68.6 | 72.1 | 95.1 | 96.6 | 97.1 |
| 4 | 92.4 | 87.0 | 86.6 | 86.5 | 92.4 | 96.8 | **97.6** | **97.9** |
| 6 | 92.8 | 87.2 | 86.8 | 86.7 | 92.8 | 96.7 | 97.4 | 97.8 |
| 8 | **92.9** | **87.4** | **87.0** | **86.9** | **92.9** | **96.9** | 97.5 | 97.8 |

Table 6: Ablation study on the sequence length T.

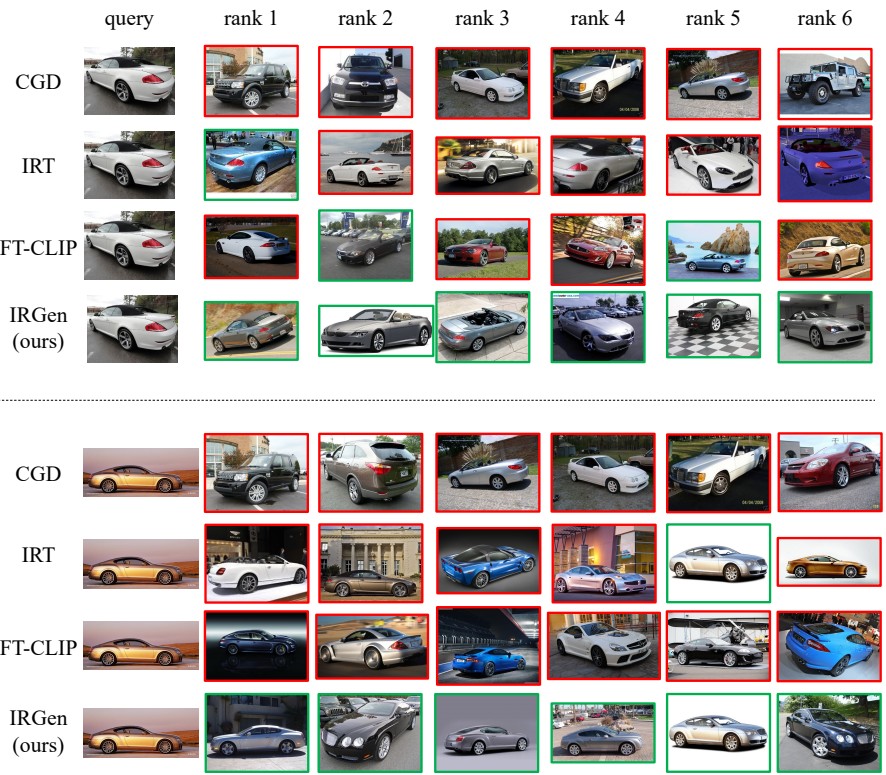

Figure 6: Examples on Cars196 dataset.Results of CGD, IRT, FT-CLIP, our IRGen are shown from top to bottom.The results of CGD, IRT, FT-CLIP are retrieved by SPANN.

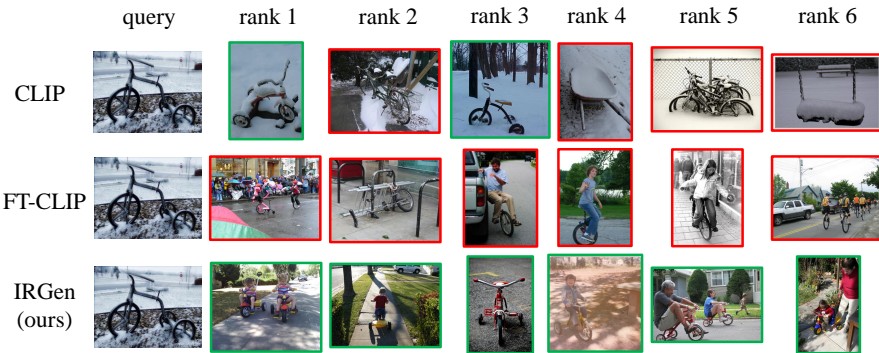

Figure 7: Examples on ImageNet dataset.Results of CLIP, FT-CLIP, our IRGen are shown from top to bottom. The results of CLIP, FT-CLIP are retrieved by SPANN.

## C  QUALITATIVE RETRIEVAL RESULTS

In this section, we provide several retrieval examples that showcase the performance of our approach compared to baselines. The retrieval results on In-shop Clothes, Cars196, and ImageNet using different methods are depicted in Figure 8, Figure 6, and Figure 7, respectively. Correctly retrieved images are highlighted with green borders, while incorrectly retrieved ones are marked with red borders. Upon examining the results presented in these figures, it becomes evident that our proposed method performs exceptionally well and is capable of handling even the most challenging examples.

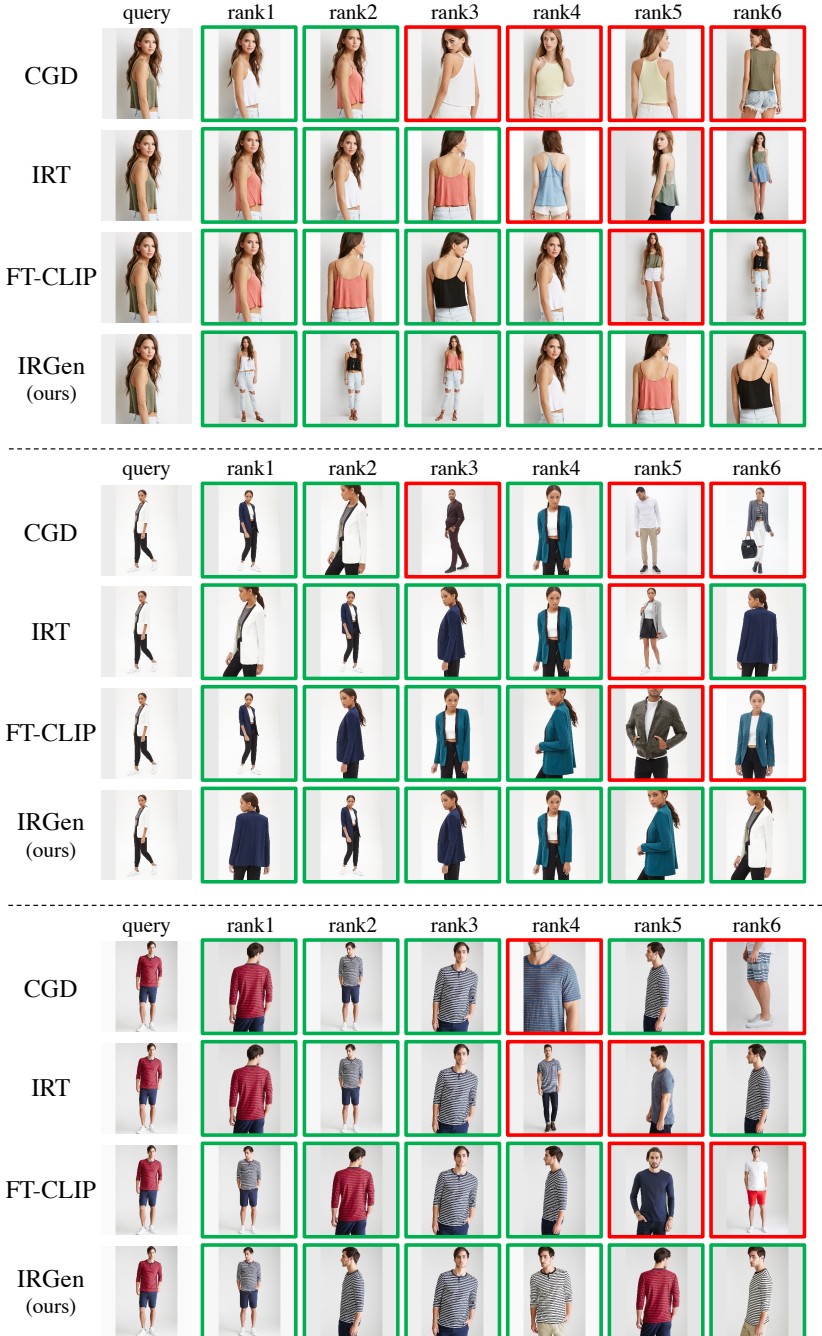

Figure 8: Examples on In-shop Clothes dataset.Results of CGD, IRT, FT-CLIP, our IRGen are shown from top to bottom.The results of CGD, IRT, FT-CLIP are retrieved by SPANN.

