# OpenReview forum: "IRGen: Generative Modeling for Image Retrieval"
_ICLR.cc/2024/Conference — Submitted to ICLR 2024_

### Official Review · Reviewer_rUYC · 2023-10-27

**Soundness:** 2 fair
**Presentation:** 1 poor
**Contribution:** 1 poor
**Rating:** 5
**Confidence:** 3

**Summary:**

The paper poses image retrieval as a form of generative modelling in order to allow the use of common transformer architectures for search in an end-to-end fashion. The proposed IRGen gets a query image as input and outputs identifiers that correspond to the nearest neighbors of the query in a given image database. This effectively turns search into a problem solved by a large transformer encoder-decoder
combines the two classical steps of image retrieval (feature extraction and indexing/search) into one module, that has to be trained for a given database.

They introduce a "semantic image tokenizer" that transforms an image into a sequence of tokens by only taking into account a global image feature from the CLS token. They use resdual quantization and alternative optimization to

**Strengths:**

The paper deals with an interesting formulation of image indexing as a sequence problem over quantized representations.  A differentiable index is used and therefore one can finetune the model end to end. Despite there are a number of unclear parts regarding the papers actual contribution and evaluation protocol, the fact that this approach works well is interesting.

**Weaknesses:**

1) The paper is written in a way that is hard to follow and understand, both the method and the contributions. Figure 1 is ambiguous or wrong.

2) A lot of relevant work is missing and therefore so is proper discussion on the method's relation to other works, especially in the area of deep quantization and hashing. Some example are listed below, many more can be found through the works listed.

3) The paper's contributions are unclear and while there are 2 components introduced, they are only tested together and it is hard to understand the contribution of each. Specifically, The tokenizer is pretrained on top of a large visual encoder (ViT from CLIP) independently from the index via alternative optimization. This module gives a set of codes that can be used for retrieval. The performance of those codes (seen as quantization codes) is not evaluated, while a clear discussion on technical differences and contributions to other residual quantization methods or other recent works on deep quantization or hashing is missing. It is also unclear how much the fact the quantized codes are a sequence matters, versus eg using simple PQ codes that have a much easier to optimize objective. Note that the method here uses annotated data from the downstream task, therefore the fair comparisson would be vs supervised quantization methods.

4) The large AR model on top acts as a differentiable index. There is no proper discussion on how this compares to other differentiable indexes from a technical perspective. The only difference mentioned to other recent differentiable indexing methods in the end of sec 4 is that the input features (document identifiers) are obtained in a different way, but this has nothing to do with the actual differentiable index modeling part.

5) The training objective is not much different from pairwise objectives on top of quantized codes used in quantization/hashing works. Although one has to pass through the index, only image pair codes are used for computing the loss. This is why beam search is needed at test time,  ie there is a clear train test discrepancy here, same as training deep hashing. I cannot see how this is more "end2end", as the objective of the loss is not a full database ranking objective. Also note that beam search is a complex data structure that keeps multiple paths and also "a prefix tree containing valid codes". One can say maybe that beam search is in itself as complex as any other indexing structure.

6) Experimental validation is generally lacking. As mentioned above, the contrubution of the two modules separately shoudl be studied. Also, reporting the precision and recall metrics in isolation is not the best for image retrieval. Results with mAP (or at least mAP100) for all datasets should be presented. Also results on classic retrieval datasets like Oxford and PAris [ir1] are missing.

7) inference speed experiments lacking and misleading: The proposed method has a large computational overhead, ie a large transformer decoder instead of a simple index. Basically, a comparisson versus FT-CLIP + some indexing is needed, but I can only see the proposed approach to be possibly even orders of magintude slower: excluding feature extraction with CLIP (the proposed method uses CLIP as the encoder to give to the tokenizer), the proposed method needs at test timeto also 1) to tokenize, 2) pass through a large ViT decoder  (12
transformer decoder block with dimension 768 for the small datasets, 24 layers for IamgeNet) and 3) beam search. Instead, Ft-CLIP with a basic IVFPQ index only needs to search the index and this is really really fast.

8) The model is highly task specific: a different trained AR model/index is needed for every database. Also it is unclear how such an index can handle changes in the database

Some sample missing references:

[dq1] YK Jang, NI Cho  Self-supervised Product Quantization for Deep Unsupervised Image Retrieval. ICCV 2021

[dq2] Yue Cao, Mingsheng Long, Jianmin Wang, Han Zhu, and Qingfu Wen. Deep quantization network for efficient image retrieval. In AAAI, 2016.

[dq3] Young Kyun Jang and Nam Ik Cho. Generalized product quantization network for semi-supervised image retrieval. In CVPR, 2020

[dq4] Benjamin Klein and Lior Wolf. End-to-end supervised product quantization for image search and retrieval. In CVPR,
pages 5041–5050, 2019.

[dh1]  Kamran Ghasedi Dizaji, Feng Zheng, Najmeh Sadoughi, Yanhua Yang, Cheng Deng, and Heng Huang. Unsupervised deep generative adversarial hashing network. In CVPR, pages 3664–3673, 2018

[dh2] Jingkuan Song. Binary generative adversarial networks for image retrieval

[ir1] Radenović, Filip, Giorgos Tolias, and Ondřej Chum. "CNN image retrieval learns from BoW: Unsupervised fine-tuning with hard examples." In ECCV 2016

**Questions:**

Please respond to the comments above.

Q1: In Fig 1 shouldnt the tokenizer get the ouput of a visual encoder as input? ie f_cls? where is that? Is "Semantic Image Tokenizer" a separate encoder than the "transformer encoder" in Fig 1?
Q2: For the same image encoder f_cls, what is the retrieval performance of the proposed tokenizer vs other PQ/hashing tokenizers?

---

> ### Author Response · Authors · 2023-11-22
> **Rebuttal by Authors, Part 1/3**
>
> Thank you for taking the time to review our paper, but there exists serious misunderstanding about our paper.
>
> First, we address the points you mentioned in the weaknesses part.
>
> 1. Figure 1 illustrates our AR training methodology that given an input image, we feed it to our transformer (encoder and decoder) and the output is supervised by the image identifier that is obtained from the query's nearest neighbor through semantic image tokenizer (which is trained before AR model). To address any misunderstandings, we kindly request more specific details regarding your concerns.
> We itemize our contribution as follows for clarity:
> (1) To the best of our knowledge, our work represents the first effort in employing a sequence-to-sequence generative model to address image retrieval in a fully end-to-end manner that directly generates the nearest neighbor's identifier givent the query.
> (2) We introduce a semantic image tokenizer explicitly designed for image retrieval to obtain image identifiers, facilitating autoregressive training. It is essential to note that this tokenizer differs significantly from previous quantization models, as our identifier solely represents the image and is not employed for distance computation, as is the case in quantization.
> (3) Extensive experiments show that our approach exhibits substantial improvements in results, achieving new state-of-the-art results. We have demonstrated its scalability to million-scale datasets and provided throughput number, showcasing its near-real-time performance.
>
> 2. We provide a more in-depth discussion comparing our approach with prior joint learning methods. In the existing literature, early methods predominantly focused on three distinct aspects: embedding learning for enhancing search accuracy, embedding compression (quantization/hashing) for reducing memory cost, and clustering (often involving quantization) or neighborhood graphs for improving search efficiency. However, as each of these components is essential for a large-scale retrieval system, integrating individual efforts may result in suboptimal solutions. Consequently, there has been a growing interest in studying joint learning. However, all these prior works (please refer to our general response for references to prior works) focus on end-to-end training of encoding and compression together while overlooking the indexing part (meaning non-exhaustive search), which is challenging to formulate as differentiable. They leverage quantization (compression index) for ANN search (with distance approximation) but still rely on linear scan in their experiments. In contrast, our approach reframes retrieval as a completely novel end-to-end task, eliminating explicit distinctions between embedding, compression, and indexing. Our method involves offline storage of image identifiers acquired from the proposed tokenizer. Notably, this differs conceptually from quantization, as we do not use the tokenized identifier for distance comparison as in quantization. Retrieving a query's nearest neighbor in our method only involves passing the query through the autoregressive (AR) model. We will incorporate this discussion into the final paper.
>
>
>
>
>
> 3. There is a serious misunderstanding that needs clarification. The codes generated by our system are intended to serve as image identifiers, not quantization codes as suggested in your comment. It's important to note that these codes are not utilized for distance computation, as is the case with quantization. Moreover, our model does not rely on the codes or codebooks during the retrieval process. Instead, it dynamically generates the codes through the autoregressive decoder based on the provided query. This methodology diverges substantially from conventional deep quantization methods.
> Then we address the concerns you raised in the comment.
> (1) We have itemized our contribution in the response above.
> (2) To assess the efficacy of the semantic image tokenizer, we have provided ablations in Table 5 of the paper, which involves ablating the image identifier using random IDs or hierarchical k-means IDs. For AR decoder, we have demonstrated its superior performance compared to FT-CLIP with linear scan, as indicated in Table 1 and Table 2 of the paper. Furthermore, we add additional comparison with a baseline that directly utilizes codes from the semantic tokenizer for image retrieval, akin to how conventional quantization performs retrieval. As anticipated, given the information loss inherent in quantization, this method is naturally inferior to linear scan and, consequently, to our proposed method.
>
> | Method \ Inshop   | Prec@1   | Prec@10  | Prec@20  | Prec@30  |
> |:-------- |:--------:| --------:| --------:| --------:|
> | IRGen     |  92.4    |  87.0    |  86.6    |  86.5    |
> | RQ-ADC|     62.5    |  48.0  |  44.8   |  43.3  |
>
> (3) The distinctions between our method and previous approaches, such as deep quantization, have been elucidated in the response above.

---

> ### Author Response · Authors · 2023-11-22
> **Rebuttal by Authors, Part 2/3**
>
> (4) We add comparison to a baseline that uses PQ for tokenizer instead of RQ. The superior performance demonstrates that sequence matters in our autoregressive training.
>
> | Model \ Inshop   | Prec@1   | Prec@10  | Prec@20  | Prec@30  |
> |:-------- |:--------:| --------:| --------:| --------:|
> | IRGen (proposed tokenizer)     |  92.4    |  87.0    |  86.6    |  86.5    |
> | IRGen (PQ tokenizer)      |  73.8    |  66.1    |  61.8    |  59.2    |
>
> (5) We compare with recent supervised deep quantization methods on ImageNet, including the state-of-the-art methods ADSVQ and DPQ. Notably, these methods use 100 classes from ImageNet, as indicated in their original papers. For fair comparison, we reran our model for 100 classes, whereas the results in our paper are based on the full 1000 categories of ImageNet. The table below illustrates the superiority of our approach.
>
> |  Method  |   mAP@1000 for 32 bits (equivalent to the length of our identifier)  |
> | --------:| ------------:|
> |   IRGen     |     93.52     |
> |DPQ [1] | 87.70 |
> |   ADSVQ [2]  |     75.21     |
> |   DTQ [3]   |     68.12     |
> |ADSH [4] | 63.36|
> |HashNet [5]| 58.71|
>
>
> [1] Klein, Benjamin, and Lior Wolf. "End-to-end supervised product quantization for image search and retrieval." Proceedings of the IEEE/CVF Conference on Computer Vision and Pattern Recognition. 2019.
>
> [2] Zhou, Chang, Lai Man Po, and Weifeng Ou. "Angular deep supervised vector quantization for image retrieval." IEEE Transactions on Neural Networks and Learning Systems 33.4 (2020): 1638-1649.
>
> [3] Liu, Bin, et al. "Deep triplet quantization." Proceedings of the 26th ACM international conference on Multimedia. 2018.
>
> [4] Liu, Haomiao, et al. "Deep supervised hashing for fast image retrieval." Proceedings of the IEEE conference on computer vision and pattern recognition. 2016.
>
> [5] Cao, Zhangjie, et al. "Hashnet: Deep learning to hash by continuation." Proceedings of the IEEE international conference on computer vision. 2017.
>
>
>
> 4. Differentiable indexing in our approach means non-exhaustive search, whereas deep quantization methods use differentiable index for embedding compression. Our focus on image retrieval marks the first effort. We introduce a novel semantic tokenizer specifically designed for image retrieval. We add a direct comparison using NCI's hierarchical k-means tokenizer versus ours, emphasizing the effectiveness of our proposed semantic tokenizer.
>
> | Method \ Inshop   | Prec@1   | Prec@10  | Prec@20  |
> |:-------- |:--------:| --------:| --------:|
> | IRGen     |  92.4    |  87.0    |  86.6    |
> | NCI (HKM)  | 89.0 |81.6 |79.8 |
>
> 5. There is a serious misunderstanding. Our model does not use image pair codes for computing the loss, as indicated in your comment. Instead, we aim to generate the codes for the nearest neighbor corresponding to the given input. The distinctions between our model and previous deep quantization methods have been elaborated in the response above. Our more end-to-end property stems from that we directly find the nearest neighbor through the learned generative model, compared to deep quantization that needs to calculate distance between query and each database's codes. When only the top-1 sample is required, beam search is unnecessary, and generation can be performed by straightforwardly selecting the identifier with the highest probability, ensuring no train-test discrepancy. However, for top-K retrieval, we leverage beam search with a configured beam size of K. In this scenario, a train-test discrepancy emerges. One potential solution is the further integration of beam search into the training process, a direction that is worth pursuing.
>
> 6. It appears that certain experiments presented in the paper may have been overlooked. The contributions of the two modules have been examined, and please refer to point (2) in the 3rd response. We have conducted a mAP comparison on two million-scale datasets, ImageNet and Places365, as in Table 3 of the paper. Regarding the remaining three datasets in Table 1 and 2, a direct mAP or mAP@100 comparison is not reasonable due to the limited number of ground truth samples, all less than 100 for each query. It is important to note that previous works, including the compared baselines, only report recall@K metric in their experiments. To provide a comprehensive comparison, we reported precision@K metric in our paper. Furthermore, we have evaluated on three widely used benchmarks and two million-scale datasets. The classical retrieval datasets, Oxford and Paris, are relatively small scale, with 5062 images for Oxford and 6412 images for Paris. These datasets are commonly employed for zero-shot retrieval scenario, particularly for pre-training models. Notwithstanding, we add comparison to Paris, which consistently validates the effectiveness of our proposed method.
>
>
> | Method \ RParis |   mAP for M  | mAP for H |
> | :------| ------: | -------: |
> |   IRGen   | 75.7 |40.4 |
> | FT-CLIP| 71.4 |27.1 |

---

> > ### Author Response · Authors · 2023-11-22
> > **Rebuttal by Authors, Part 3/3**
> >
> > 7. There is a serious misunderstanding that needs clarification. At test time, our model does not use the tokenizer, as this step is exclusive to database images and can be executed offline. The query only undergoes a transformer encoder and decoder process to search for its nearest neighbor. Similarly, other baselines like FT-CLIP and IRT also involve passing the query through a transformer encoder to obtain the query embedding, resulting in same time cost for the encoder at test time. Secondly, despite our decoder being either 12-layer or 24-layer, its efficiency stands out due to the notably short sequence length. Lastly, beam search is conducted concurrently with the decoding process. We have provided throughput comparison on Inshop dataset in Figure 5 in the paper. Here we provide a comprehensive comparison focusing on the ImageNet dataset. It is crucial to emphasize that a retrieval system's storage considerations extend beyond just the model; it must also store features for all database vectors, constituting a substantial portion of storage requirements. In terms of inference cost, our CPU inference cost is 0.30s per image, which is not orders of magnitude larger compared to FT-CLIP with linear scan. Although FT-CLIP with SPANN lowers the inference cost, it comes at the expense of a substantial drop in performance. Our GPU inference cost, tested on a Tesla P100 PCIe 16GB GPU, is 0.037s/image, which is near-real-time, considering practical retrieval systems at the order of 10 milliseconds. Moreover, our approach demonstrates competitive performance and significantly reduces the memory cost. We anticipate that the prevalent reliance on GPUs for generative models will swiftly lead to newer GPU generations with increased processing power, more efficient memory management, and improved hardware support, thereby enhancing the inference speed of our autoregressive retrieval. Additionally, our model, being GPT-based, can leverage rapidly developed acceleration techniques designed for GPT.
> >
> > |  Method | Model (#Params) |  Database storage  |
> > | -------:| ----------------- |  --------------:|
> > |      IRGen | ViT-base encoder (86M)+ 24-layer decoder (307M)  | 4.9MB： 4 bytes (id)/image with 1281167 samples    |
> > | FT-CLIP | ViT-base (86M)    | 3.67GB：  768x4 bytes (feature)/image with 1281167 samples  |
> >
> >
> > |  Method | Search         | Inference cost | mAP@100    |
> > | -------:| -------------- | --------------:| --- |
> > |      IRGen | Autoregressive |   0.037s/image |   76.0  |
> > | FT-CLIP | Linear scan (SPANN)   |   0.14s/image (0.007s/image) |   77.0 (65.5)  |
> >
> > 8. Certain experiments presented in our study seem to have been overlooked. We have conducted experiments to demonstrate the capability to handle distribution change by adding new, unseen data within the same class, as shown in Table 4 in the paper. In scenarios where more new data come in and the distribution of gallery data has changed drastically, all existing data-dependent retrieval methods, including embedding model in image retrieval as well as dense retrieval methods in document retrieval, have to re-train the model and re-construct the data index. Essentially, dynamic updates only support a small number of new data that do not substantially alter the distribution.  Nevertheless, it's important to highlight that data-dependent models continue to be preferred due to their superior performance. In future work we may extend IRGen to an adaptive structure capable of capturing the evolving distribution of new data during training.
> >
> >
> > Then we answer the questions below.
> >
> > 1. f_cls comes from the tokenizer instead of the visual encoder. They are separate models with different weights.
> >
> > 2. We add experiment comparison to a baseline that uses PQ for tokenizer instead of ours. The superior performance observed underscores the effectiveness and superiority of our tokenizer.
> >
> > | Model \ Inshop   | Prec@1   | Prec@10  | Prec@20  | Prec@30  |
> > |:-------- |:--------:| --------:| --------:| --------:|
> > | IRGen (proposed tokenizer)     |  92.4    |  87.0    |  86.6    |  86.5    |
> > | IRGen (PQ tokenizer)      |  73.8    |  66.1    |  61.8    |  59.2    |

---

> > > ### Comment · Reviewer_rUYC · 2023-11-23
> > > **Thank you for detailed answers**
> > >
> > > I want to thank the reviewers for detailed answers and let me apologize for any misunderstandings on my side. I now see how there exist conceptual differences with quantization and get the novelty aspects better.
> > >
> > > I still think the paper is not clearly written, and for Figure 1 the authors did nto respond to my question: does  the tokenizer have a visual encoder inside? If so, does it  share weights with the "transformer encoder" in Fig 1?
> > >
> > > I also thank the authors for providing comparisons to deep quantization methods. However, it is unclear to me that they are fair - specifically: Is the tokenizer in your method _pretrained_ on top of a large visual encoder from CLIP? If so, this gives an unfair advantage to your method when you compare to other hashing/PQ methods (that I would assume only train on the ImageNet dataset - plz correct me if I am wrong).
> > >
> > > In fact, I would appreciate if the authors list here _all the data_ that their method (tokenizer and autoencoder) was trained on. This includes any pretrained models they might have used as initialization.
> > >
> > > It is also unclear to me if the authors uploaded an updated pdf - i believe this to be important given the fact that the paper is not clearly written and the authors added related works and comparissons in their answers. If the authors did, I  apologize and would encourage them to color all changes so that one can easily spot new additions/rephrasings/related work.
> > >
> > > I will increase my score to borderline reject; Waiting for a response from the authors to my questions above.

---

### Official Review · Reviewer_obPk · 2023-10-30

**Soundness:** 3 good
**Presentation:** 2 fair
**Contribution:** 2 fair
**Rating:** 5
**Confidence:** 2

**Summary:**

The paper introduces an image retrieval method built upon semantic image tokenization. Recognizing the limitations of popular tokenization techniques like VQ-VAE and RQ-VAE for retrieval tasks, the authors propose a more efficient approach that emphasizes global feature extraction from class tokens, reducing sequence lengths and emphasizing high-level semantic information. To enhance semantic representation, they also incorporate classification loss training. In tandem, an autoregressive encoder-decoder architecture is employed that decouples input embedding from discrete code generation, focusing on understanding semantic relationships between image pairs. During the inference stage, beam search is used to efficiently find top-K matches for a given query image. The model's end-to-end design ensures efficient retrieval and offers a novel perspective compared to traditional approximate nearest neighbor search techniques.

**Strengths:**

1. The paper introduces an encoder-decoder architecture that decouples input embedding from discrete code generation. This design, focusing on understanding the semantic relationships between image pairs, offers a more flexible and adaptable framework, potentially making the model more robust and applicable across various datasets and retrieval scenarios.

2. The model's design places a significant emphasis on capturing high-level semantic information. By incorporating classification loss training on both original and reconstructed embeddings, the proposed approach ensures that the retrieved images are semantically relevant to the query, bridging the gap between low-level details and meaningful content.

3. The paper acknowledges the inefficiencies of traditional tokenization techniques for image retrieval and introduces an approach that reduces sequence lengths. By focusing on global feature extraction from class tokens, the model offers a more streamlined and efficient representation, especially suited for retrieval tasks.

**Weaknesses:**

1. While emphasizing global feature extraction from class tokens might improve efficiency, there's a risk of overlooking crucial spatial information present in other parts of the image, possibly leading to incomplete or less accurate retrieval results.

2. The paper proposes the use of beam search for efficient top-K retrieval, but this method can be computationally intensive, especially for large image databases. Additionally, validating each generated image identifier can be a time-consuming process, even with their proposed prefix tree optimization.

3. The approach, especially the beam search with constraints, may face scalability issues when dealing with vast and diverse image datasets. As databases grow, the efficiency and accuracy of the method may be challenged.

4. While the method aims to capture high-level semantics, it might not generalize well across diverse datasets with varying characteristics. The paper does not address how the model would adapt to datasets with vastly different semantic structures or image content.

5. There seems to be a limited discussion on how this method performs compared to other state-of-the-art techniques. Without comprehensive comparative studies, it's challenging to ascertain the model's effectiveness and superiority in real-world scenarios.

**Questions:**

1. How do you ensure that the emphasis on global feature extraction from class tokens doesn't compromise the finer spatial details of the image, which could be vital for certain retrieval scenarios?

2. Given the complexities of beam search, especially for large image databases, what specific optimizations have you implemented to ensure real-time or near-real-time retrieval performances?

3. How does the proposed model handle noisy or imperfect image datasets? Are there specific preprocessing steps or augmentations recommended to enhance retrieval accuracy?

4. How adaptable is the autoregressive encoder-decoder architecture to other types of multimedia content, such as videos or audio? Are there potential modifications or extensions to the proposed method for such content?

5. Given the model's dependency on class tokens, how does it handle images that may not fit neatly into predefined classes or those that belong to multiple overlapping classes?

6. You have mentioned using classification loss for training. Were other loss functions explored during the development? If so, how did they impact the model's performance?

---

> ### Author Response · Authors · 2023-11-22
> **Rebuttal by Authors, Part 1/2**
>
> We appreciate the time you took to review our paper.
>
> First, we address the five points you mentioned in the weaknesses part.
>
> 1. Image retrieval is a high-level vision task, akin to extreme classification problem, reliant on a global understanding of the image, where global features are pivotal. Our method proves effective not only in general retrieval, such as ImageNet, but also excels in fine-grained tasks like Inshop clothes retrieval, CUB retrieval, and car retrieval, showcasing its ability to capture necessary details for fine-grained search using global features. Recognizing scenarios where similarity is defined over small image areas requiring spatial information, we suggest a solution involving a pre-processing step that detects these important regions. For instance, person re-identification is such a case utilizing detection before search. Alternatively, incorporating both global and local features ensures a comprehensive image representation but may entail complex models with increased computational demands. Striking a balance between efficiency and efficacy is a longstanding challenge in retrieval. This paper has successfully present a novel approach that is conceptually different from previous joint learning methods, and demonstrate state-of-the-art performance on widely used benchmarks with near-real-time inference speed.
>
> 2. First, the proposed prefix tree only contains valid codes and thus eliminates the need of validating each generated image identifier, largely enhancing the inference speed. We then provide a comprehensive comparison of model capacity and inference cost in the table below, focusing on the million-scale ImageNet dataset. It is crucial to emphasize that a retrieval system's storage considerations extend beyond just the model; it must also store features for all database vectors, constituting a substantial portion of storage requirements, especially in the context of a large database. While compression can alleviate this concern, the performance is severely compromised due to quantization. In terms of inference cost, our CPU inference cost is 0.30s per image, which is not orders of magnitude larger compared to FT-CLIP with linear scan. Although FT-CLIP with SPANN lowers the inference cost, it comes at the expense of a substantial drop in performance. Our GPU inference cost, tested on a Tesla P100 PCIe 16GB GPU, is 0.037s/image, which is near-real-time, considering practical retrieval times at the order of 10 milliseconds. Moreover, our approach demonstrates competitive performance compared to FT-CLIP with linear scan, and significantly reduces the memory cost. For inference speed, we anticipate that the prevalent reliance on GPUs for generative models will swiftly lead to newer GPU generations with increased processing power, more efficient memory management, and improved hardware support, thereby enhancing the inference speed of our autoregressive retrieval. Additionally, our model, being GPT-based, can leverage rapidly developed acceleration techniques designed for GPT.
>
> |  Method | Model (#Params) |  Database storage  |
> | -------:| ----------------- |  --------------:|
> |      IRGen | ViT-base encoder (86M)+ 24-layer decoder (307M)  | 4.9MB： 4 bytes (id)/image with 1281167 samples    |
> | FT-CLIP | ViT-base (86M)    | 3.67GB：  768x4 bytes (feature)/image with 1281167 samples  |
>
>
> |  Method | Search         | Inference cost | mAP@100    |
> | -------:| -------------- | --------------:| --- |
> |      IRGen | Autoregressive |   0.037s/image |   76.0  |
> | FT-CLIP | Linear scan (SPANN)   |   0.14s/image (0.007s/image) |   77.0 (65.5)  |
>
> 3. Our current objective is to showcase the effectiveness of the proposed generative based model. We have conducted experiments on three widely used retrieval benchmarks, and show the scalability on other two wellknown million-scale datasets. Further scaling to handle billion-scale datasets poses a considerable challenge due to the substantially increased demand for resources, including storage and computation, which currently exceeds our available capacity.  In future work, we may address billion-scale scenario with a hybrid solution where billions of images can be clustered into millions of semantic clusters, and our proposed model IRGen can then be utilized to retrieve the relevant cluster identifiers. This approach would enable us to leverage current success in million-scale datasets and handle billion-scale datasets in a more efficient manner.

---

> > ### Author Response · Authors · 2023-11-22
> > **Rebuttal by Authors, Part 2/2**
> >
> > 4. In scenarios where more new data come in with vastly different semantic structure, leading to the distribution of gallery data change drastically, all existing data-dependent retrieval methods, including embedding model in image retrieval as well as dense retrieval methods in document retrieval, have to re-train the model and re-construct the data index. Essentially, generalization (or dynamic updates) only supports a small number of new data that do not substantially alter the distribution. Our experiments have demonstrated this case in Table 4 in the paper, showcasing the model's adaptability to minor distribution changes. Nevertheless, it's important to highlight that data-dependent models continue to be preferred due to their superior performance. In future we may extend IRGen to an adaptive structure capable of capturing the evolving distribution of new data during training.
> >
> > 5. In our experiments, we extensively compared our approach with state-of-the-art techniques. The dataset leaderboards, accessible at the paperwithcode website, highlight CGD as the leader. Our paper has provided a detailed comparison of our approach with it, demonstrating a significant improvement margin. We will make it more clear in the final revision.
> >
> > Then we answer the questions below.
> >
> > 1. Already presented in the weakness part. Moreover, as we fine-tune the CLIP image encoder concurrently with our model, it is plausible that the acquired features will dynamically adapt to capture intricate details essential for effective similarity search. This joint optimization process enhances the model's ability to discern subtle nuances, as verified by the fine-grained retrieval.
> >
> > 2. We introduce a prefix tree that eliminates nodes lacking children, ensuring that only valid image IDs can be reached through the prefix tree. This design significantly accelerates beam search, as there is no need to check the validity of each generated image identifier. A detailed comparison of model capacity and inference cost is provided in the weaknesses part, demonstrating that our approach achieves near-real-time retrieval performance.
> >
> > 3. We would like to point out that to demonstrate the effectiveness of the proposed framework, our approach, same to all previous methods, evaluate the retrieval performance on the widely used benchmarks tailored to minimize the presence of noisy images. In real-world scenarios, however, noise and imperfections are inevitable. Two strategies can be employed. 1) Data Filtering: utilize a designed mechanism for outlier removal or employ a simple classifier trained on labeled data (good quality vs. low quality) to filter extreme noise from the dataset. 2) Model regularization such as data augmentation (rotation, flip, etc) to enhance the model robustness, and dropout regularization to prevent overfitting.
> >
> > 4. We would like to break the extension into two scenarios, one is extension to retrieval in another datatype such as videos or audio, and the other is extension to multi-modal retrieval in multi-modal data. For the first case, the autoregressive encoder-decoder architecture can be modified for alternative data types. Both the AR model and the semantic image tokenizer would need adjustments to accommodate video or audio inputs, presenting potential challenges in the process. In the second, more intriguing scenario, an additional operation is necessary to align the feature space from multiple modalities. We plan to delve into this aspect in future research.
> >
> > 5. For the first question, we've highlighted in the weaknesses part that all existing data-dependent retrieval methods require re-training and re-construction of the data index to handle new images with significantly different semantic structures. To address this, we envision potential modifications to IRGen, transforming it into an adaptive structure capable of capturing the evolving distribution of new data during training in the future. Regarding the second question, when dealing with images belonging to multiple overlapping classes, the primary challenge is defining similarity between two images. Typically, a score reflecting the degree of overlap between two images can be assigned. This score indicates that if the two images share more class labels, their similarity score is higher. Once similarity scores are determined, we can adapt the training in the AR model to utilize triplet ranking loss.
> >
> > 6. The classification loss is used for our image identifier training. In addition to that, we propose to utilize M levels of partial reconstruction loss, which is shown in Equation (3) in the paper. We add ablation experiments to show how it impact the model's performance.
> >
> > | Model \ Inshop   | Prec@1   | Prec@10  | Prec@20  | Prec@30  |
> > |:-------- |:--------:| --------:| --------:| --------:|
> > | Full loss   |  92.4    |  87.0    |  86.6    |  86.5    |
> > | Only classification loss     |  92.1    | 85.1    |  83.4   |  82.5    |

---

> ### Comment · Reviewer_obPk · 2023-11-23
> **Thanks for the rebuttal**
>
> I thank the authors for preparing the rebuttal and answering my comments. After reading authors' reply and other reviews, I am keeping my original rating. Specifically, the advantage of prefix tree and the formulation is still not clear to me. Additionally, there are other concerns requiring the paper to be updated to see how the updated manuscript would look like. Furthermore, the model diagram doesn't contain all the proposed components, which in my opinion is necessary.

---

### Official Review · Reviewer_8qfY · 2023-10-30

**Soundness:** 4 excellent
**Presentation:** 4 excellent
**Contribution:** 4 excellent
**Rating:** 6
**Confidence:** 4

**Summary:**

This paper proposes a novel approach for image retrieval using generative modeling -- IRGen. IRGen is a sequence-to-sequence model that, given a provided query image, directly generates "identifiers" corresponding to the query’s nearest neighbors. Specifically, the model takes a query image as input and autoregressively predicts discrete visual tokens, which are considered as the identifier of an image. These discrete visual tokens are learned through classification loss, the global features of an image is tokenized through residual quantization. Once the semantic image tokenizer is trained, then a decoder is learned to predict the identifier of query's nearest neighbor through autoregressive way. In summary, this paper propose a novel approach to tokenize an image into semantic identifiers and achieves state-of-the-art over conventional methods.

**Strengths:**

originality:  the paper proposes a novel approach for image retrieval using generative modeling. The semantic image tokenizer with residual quantization is elegant and the generative modeling way is quite interesting. The whole framework looks pretty straightforward and simple.

quality: the paper is technically sound. Extensive experiments were performed.

clarity: the paper is well-written and well-organized.

significance: the paper address image retrieval with generative modeling which is quite an interesting way, showing the possibility of using generative method to obtain codes for retrieval. Hence this paper can inspire many future works, such as how to learn a code that can do both generation and retrieval, how to speed up the search (e.g., with hash codes/without beam search).

**Weaknesses:**

- the description of how the beam search is a little bit confusing to me. I couldn't fully understand how it is done.
- retrieval with autoregressive means the retrieval may not enjoy the benefit of maximum inner-product search
- as the current retrieval is done with GPU, if using CPU, the decoder process will be significantly slowed down and may affect the retrieval speed

**Questions:**

1. does the semantic tokenizer share the weights with the encoder?
2. seems like we can directly use the code from semantic tokenizer for image retrieval -- similar to how product quantization performs retrieval, what is the necessity of employing the decoder? What is the disadvantange/how much performance degrades if we just use the semantic tokenizer's code for retrieval?
3. what is the retrieval speed of using IRGen on CPU compared to IVFPQ/ScaNN/SPANN?
4. what if you train a normal classification model, then perform PQ to obtain the "tokens" instead of the proposed semantic tokenizer?
5. is a two-stage training process? can they be trained end-to-end?

---

> ### Author Response · Authors · 2023-11-22
> **Rebuttal by Authors, Part 1/2**
>
> Thank you for your detailed review.
>
> First, we address the three points you mentioned in the weaknesses part.
>
> 1. The procedure begins by defining a beam size, denoted as K in our case. Initially, the beam comprises the start-of-sequence token, and the algorithm extends the current set of candidate sequences by considering the next potential tokens in the sequence. For each candidate sequence, the algorithm outputs probabilities for multiple potential next tokens and adds them to the beam. The expanded set of sequences is then scored based on the probabilities assigned by the model. The top-k sequences with the highest scores are retained, while the others are discarded. This selection process, which can be implemented with a priority queue in linear time with the candidate list size, is repeated until the end-of-sequence token is encountered. In the case of beam search with a prefix tree, the search process is constrained to consider only valid next tokens, as determined by the prefix tree, rather than all the possible next tokens. This constraint ensures that the search process results in valid image identifiers corresponding to specific images in the dataset. We will provide a detailed explanation of the beam search algorithm in the Appendix.
>
> 2. We find the noted weakness a bit perplexing, particularly in the absence of specific details regarding the particular benefit that may not be enjoyed by retrieval with an autoregressive approach. Our interpretation is based on our best understanding of the context. Maximum inner-product search (MIPS) pertains to a similarity search problem where the objective is to efficiently locate the data point in a collection that maximizes the inner product with a given query vector. Nonetheless, if the ultimate goal is to retrieve similar data, our methodology allows us to formulate and address this challenge optimally in an end-to-end manner, rather than a two-step process involving initial feature embedding followed by maximum inner-product search. We acknowledge the potential scenarios where data embedding becomes imperative, such as compact feature learning for transmission purposes. In such cases, one could also treat the embedded data as input and train an autoregressive model to identify the query's nearest neighbor, bypassing the tedious tuning over  the joint integration of compression and indexing.
>
> 3. We provide a comprehensive comparison of model capacity and inference cost in the table below, focusing on the ImageNet dataset. It is crucial to emphasize that a retrieval system's storage considerations extend beyond just the model; it must also store features for all database vectors, constituting a substantial portion of storage requirements, especially in the context of a large database. In terms of inference cost, our CPU inference cost is 0.30s per image, which is not orders of magnitude larger compared to FT-CLIP with linear scan. Although FT-CLIP with SPANN lowers the inference cost, it comes at the expense of a substantial drop in performance. Our GPU inference cost, tested on a Tesla P100 PCIe 16GB GPU, is 0.037s/image, which is near-real-time, considering practical retrieval systems at the order of 10 milliseconds. Moreover, our approach demonstrates competitive performance compared to FT-CLIP with linear scan, and significantly reduces the memory cost. We anticipate that the prevalent reliance on GPUs for generative models will lead to newer GPU generations with increased processing power, more efficient memory management, and improved hardware support, thereby enhancing the inference speed of our autoregressive retrieval. Additionally, our model, being GPT-based, can leverage rapidly developed acceleration techniques designed for GPT. Our model also opens avenues for the integration of GPT in the development of more intelligent and context-aware information retrieval and generation systems.
>
> |  Method | Model (#Params) |  Database storage  |
> | -------:| ----------------- |  --------------:|
> |      IRGen | ViT-base encoder (86M)+ 24-layer decoder (307M)  | 4.9MB： 4 bytes (id)/image with 1281167 samples    |
> | FT-CLIP | ViT-base (86M)    | 3.67GB：  768x4 bytes (feature)/image with 1281167 samples  |
>
>
> |  Method | Search         | Inference cost | mAP@100    |
> | -------:| -------------- | --------------:| --- |
> |      IRGen | Autoregressive |   0.037s/image |   76.0  |
> | FT-CLIP | Linear scan (SPANN)   |   0.14s/image (0.007s/image) |   77.0 (65.5)  |

---

> > ### Author Response · Authors · 2023-11-22
> > **Rebuttal by Authors, Part 2/2**
> >
> > Then we answer the questions below.
> >
> > 1. No. They are separate models with different weights.
> >
> > 2. We add the experimental results that directly use the codes from our semantic tokenizer for image retrieval. Specifically, we use asymmetric distance computation (ADC) to approximate the distance and use linear scan for exhaustive search, similar to product quantization. The observed significant degradation in performance underscores the imperative need for our autoregressive decoder.
> >
> > | Method \ Inshop   | Prec@1   | Prec@10  | Prec@20  | Prec@30  |
> > |:-------- |:--------:| --------:| --------:| --------:|
> > | IRGen     |  92.4    |  87.0    |  86.6    |  86.5    |
> > | RQ-ADC|     62.5    |  48.0  |  44.8   |  43.3  |
> >
> > 3. Already presented in above answers to the weakness part, please refer to that.
> >
> > 4. We add the experimental comparison on Inshop dataset with the counterpart that utilizes the PQ tokenizer (performing PQ after training a normal classification model) instead of our proposed semantic tokenizer. The inferior performance of the counterpart underscores the effectiveness of our semantic tokenizer.
> >
> > | Model \ Inshop   | Prec@1   | Prec@10  | Prec@20  | Prec@30  |
> > |:-------- |:--------:| --------:| --------:| --------:|
> > | IRGen (proposed tokenizer)     |  92.4    |  87.0    |  86.6    |  86.5    |
> > | IRGen (PQ tokenizer)      |  73.8    |  66.1    |  61.8    |  59.2    |
> >
> >
> > 5. Indeed, our current process involves training a tokenizer to obtain the image identifier and subsequently training an AR model for end-to-end search. However, it's crucial to note that our model is specifically optimized with an end-to-end retrieval target, eliminating the train-inference mismatch presented in previous retrieval models. Looking forward, we envision the potential for a joint learning of both stages, further enhancing the image identifier's awareness of the retrieval process. This is an avenue we plan to explore in future work.

---

### Official Review · Reviewer_Re4A · 2023-10-31

**Soundness:** 4 excellent
**Presentation:** 3 good
**Contribution:** 2 fair
**Rating:** 6
**Confidence:** 4

**Summary:**

The paper proposes an approach which learns the encoding and indexing structure jointly for improving improving image retrieval task. It relies on VIT backend for encoding and Residual Quantization (RQ) for hierarchical semantic representation learning. The proposed solution is optimized end-to-end unlike traditional approach of creating content embeddings and running approximate nearest neighbor (ANN) search independently. The solution is evaluated across multiple dataset and multiple baseline embedding models (fine-tuned on the same dataset) coupled with different ANN search algorithms. The results indicate significantly improved the image retrieval results.

**Strengths:**

- Provides a simple, intuitive and technically sound approach, which borrows strong insights from the recent generative modeling literature and Residual Quantization concepts.
- There are multiple baselines and numerous ablations to provided for evaluations. Outperforms strong baselines.
- The paper is well motivated and written.

**Weaknesses:**

- The concept of joint training the embedding model and index structure is not entirely novel
- The approach scales well for million scale datasets, but not billions

**Questions:**

- The concept of jointly training the retrieval / embedding model and ANN index structures (or search models) is not novel. It would be great to have a more in depth review of these approaches. A quick example: "Joint Learning of Deep Retrieval Model and Product Quantization based Embedding Index", SIGIR'21. https://dl.acm.org/doi/10.1145/3404835.3462988
- Figure 4: The precision of the proposed approach increases as top-k increases. The expectation is that there should be a trade-off.
- Table 4: The generalization capability of the approach is demonstrated by taking out 5% of the ImageNet dataset during training and used them for inference/testing. However, those 5% examples are coming from the same ImageNet dataset distribution. The model already learned the semantic classes and their visual variations. Therefore this experiment does not fully demonstrate the out-of-domain capability of the proposed method. Also, it is not clear if the same treatment applied for the other baseline model compared. Need more clarity and justification for this experiment.
- There is a need for a table comparing the capacity and inference cost of the baseline models to the proposed solution in the Appendix (for Apples-to-Apples comparisons). A similar comparison is also needed for the retrieval stage. At million scale, even linear search is quite practical. It would be also great to have ablations where the model capacity is varied and the overall performance is evaluated.
- Residual Vector Quantization also uses beam search and RVQ codes may be used for efficiently knn search using prefix trees (or better with Aggregating Tree). Therefore the inference time operations are quite similar to IRGen paper except that the previous paper does not train an encoding for the search task. This could also serve as a natural baseline which would demonstrate the need for joint training of encoder and indexing/search structures. OR, after training the encoder, but could we combine and use the below paper for a more scalable (billions) search? A discussion should suffice.
Liu et al, "Generalized Residual Vector Quantization and Aggregating Tree for Large Scale Search", IEEE T on Multimedia, 2017
- Sanity check: Are the reported results for IRGen the-state-of-the-art today for the target image retrieval benchmarks? This should be stated clearly in the paper. If not, we need comparison with the SOTA method.

---

> ### Author Response · Authors · 2023-11-22
> **Rebuttal by Authors, Part 1/2**
>
> Thank you for your valuable feedback.
>
> First, we address the points you mentioned in the weaknesses part.
>
>
> 1. We offer a detailed comparison of our approach with previous joint learning methods. Early methods in the literature primarily concentrated on three key aspects: embedding learning to enhance search accuracy, embedding compression (quantization/hashing) to reduce memory cost, and clustering (often involving quantization) or neighborhood graphs to improve search efficiency. However, integrating these essential components individually may lead to suboptimal solutions for large-scale retrieval systems. As a result, there is a growing interest in the study of joint learning. However, all prior works (please refer to our general response for references to prior works) focus on end-to-end training of encoding and compression together while overlooking the indexing part (meaning non-exhaustive search), which is challenging to formulate as differentiable. They leverage quantization (compression index) for ANN search (with distance approximation) but still rely on linear scan in their experiments. In contrast, our approach reframes retrieval as a completely novel end-to-end task, eliminating explicit distinctions between embedding, compression, and indexing. Our method involves offline storage of image identifiers acquired from the proposed tokenizer. Notably, this differs conceptually from quantization, as we do not use the tokenized identifier for distance comparison as in quantization. Retrieving a query's nearest neighbor in our method only involves passing the query through the autoregressive (AR) model.
>
>
>
> 2. Thank you for acknowledging our results with million-scale datasets. Handling billion-scale datasets presents a significant challenge, given the heightened resource demands, including storage and computation, that currently surpass our available capacity. Our primary goal is to highlight the effectiveness of the proposed generative model. To demonstrate this, we conducted experiments on three retrieval benchmarks and two additional million-scale datasets. In future endeavors, we aim to address the billion-scale scenario through a hybrid solution. This involves clustering billions of images into millions of semantic clusters, allowing our model IRGen to efficiently retrieve relevant cluster identifiers. This approach would enable us to handle billion-scale datasets in a more efficient manner.
>
>
>
> Then we answer the questions below.
>
> 1. Refer to the weaknesses section for an in-depth explanation. The mentioned example also involves joint learning of encoding and compression (referred to as embedding index in the paper), relying on linear scan search in their experiments. In contrast, our approach defines retrieval as a novel end-to-end task, directly retrieving a query's nearest neighbor by passing through the AR model, without explicit modeling for embedding, compression, and indexing.
>
>
> 2. Traditional methods tend to witness a decline in precision with an increasing retrieval set size (K). While they initially achieve high precision by prioritizing the most relevant images in the top ranks, the precision drops rapidly as K grows due to the retrieval of irrelevant and confusing images. This indicates a struggle to retrieve harder positive images, often necessitating a reranking stage after the initial retrieval. In contrast, our AR decoder consistently retrieves relevant samples as the retrieval list lengthens. Our top-K performance even exhibits a slight increase with a larger K, showcasing its proficiency in handling challenging examples. This capability can be attributed to our approach of randomly sampling codes from images in the same class as the query image during training, enhancing the model's capacity to handle difficult samples. This advantage potentially mitigates the need for a post-reranking stage in our model to some extent.
>
>
> 3. In scenarios where more new data come in and the distribution of gallery data has changed drastically, all existing data-dependent retrieval methods, including embedding model in image retrieval as well as dense retrieval methods in document retrieval, have to re-train the model and re-construct the data index. Essentially, dynamic updates only support a small number of new data that do not substantially alter the distribution. Our experiments have demonstrated the capability of handling this case through adding new, unseen data within the same class, showcasing the model's adaptability to minor distribution changes. Nevertheless, it's important to highlight that data-dependent models continue to be preferred due to their superior performance. In future work, there is a potential extension of IRGen towards an adaptive structure capable of capturing evolving distribution during training.  For our experiment, the other baseline models employed the same treatment for fair comparison, and we will provide further clarification in the revised version.

---

> > ### Author Response · Authors · 2023-11-22
> > **Rebuttal by Authors, Part 2/2**
> >
> > 4. We provide a comprehensive comparison of model capacity and inference cost in the table below, focusing on the ImageNet dataset. It is crucial to emphasize that a retrieval system's storage considerations extend beyond just the model; it must also store features for all database vectors, constituting a substantial portion of storage requirements. While compression can alleviate this concern, the performance is severely compromised due to quantization. In terms of inference cost, our CPU inference cost is 0.30s per image, which is not orders of magnitude larger compared to FT-CLIP with linear scan. Although FT-CLIP with SPANN lowers the inference cost, it comes at the expense of a substantial drop in performance. Our GPU inference cost, tested on a Tesla P100 PCIe 16GB GPU, is 0.037s/image, which is near-real-time, considering practical retrieval systems at the order of 10 milliseconds. Moreover, our approach demonstrates competitive performance compared to FT-CLIP with linear scan, and significantly reduces the memory cost. For inference speed, we anticipate that the prevalent reliance on GPUs for generative models will lead to newer GPU generations with increased processing power, more efficient memory management, and improved hardware support, thereby enhancing the inference speed of our autoregressive retrieval. Additionally, our model, being GPT-based, can leverage rapidly developed acceleration techniques designed for GPT.
> > To explore the scaling law concerning model capacity, we conducted experiments using an AR model with a 12-layer decoder on the ImageNet dataset, given current time limit and computational constraints. The results in the table below indicate that our approach achieves improved performance as the model capacity increases. We intend to further investigate the performance of our method in scenarios involving larger model sizes and expanded database sizes in future work. We will add this discussion in the Appendix.
> >
> > |  Method | Model (#Params) |  Database storage  |
> > | -------:| ----------------- |  --------------:|
> > |      IRGen | ViT-base encoder (86M)+ 24-layer decoder (307M)  | 4.9MB： 4 bytes (id)/image with 1281167 samples    |
> > | FT-CLIP | ViT-base (86M)    | 3.67GB：  768x4 bytes (feature)/image with 1281167 samples  |
> >
> >
> > |  Method | Search         | Inference cost | mAP@100    |
> > | -------:| -------------- | --------------:| --- |
> > |      IRGen | Autoregressive |  0.037s/image|   76.0  |
> > | FT-CLIP | Linear scan (SPANN)   |   0.14s/image (0.007s/image) |   77.0 (65.5)  |
> >
> >
> >
> > |  Method | Model (#Params) |  mAP@100  |
> > | -------:| ----------------- |  --------------:|
> > |      IRGen | ViT-base encoder (86M)+ 24-layer decoder (307M)  |  76.0  |
> > | IRGen | ViT-base encoder (86M) + 12-layer decoder (86M)   |   74.1 |
> >
> > 5. We applied beam search to the prefix tree derived from RQ codes learned by the tokenizer. However, the performance of this approach was found to be significantly inferior to our proposed method, highlighting the importance of our end-to-end training approach. Notably, our way of accelerating beam search using a prefix tree containing only valid codes aligns with the concept of aggregating tree in the referenced work. This involves merging nodes along a path when there are no branches, effectively eliminating nodes lacking children (invalid codes in our case). Such an approach markedly reduces computational overhead. This technical design is pivotal for enhancing beam search efficiency, particularly advantageous in billion-scale search scenarios. We will incorporate this reference in the final paper.
> >
> >
> > | Method \ ImageNet  | mAP@100  |
> > |:-------- |:--------:|
> > | IRGen|  76.0   |
> > | RQ prefix-tree|  56.7  |
> >
> >
> > 6. Yes, our results achieve state-of-the-art performance for the benchmarks. We will explicitly emphasize this in the paper.

---

> > > ### Comment · Reviewer_Re4A · 2023-11-23
> > >
> > > Thanks to the authors for the detailed comments. I am in favor of keeping the current rating after reading other reviews.
> > >
> > > One side comment (which did not impact the final rating) is that I am quite surprised to see the large gap between RQ prefix-tree and IRGen comparison (MAP 56.7 vs 76). Not clear how to interpret and explain. There are also various ways to improve RQ prefix-tree solution with additional probes identified by beam search. Without these tricks, RQ prefix-tree might perform worse. For your information.

---

> > > > ### Author Response · Authors · 2023-11-23
> > > >
> > > > Thank you for your quick response.
> > > >
> > > >  Regarding the RQ prefix-tree, we employ an equivalent number of probes as our beam size (i.e., K in Top-K precision) for a fair comparison. It is crucial to highlight that each leaf node in the RQ prefix-tree typically contains more than one sample. Consequently, samples associated with the same leaf node share the same distance from the query. In contrast, our autoregressive (AR) model has the ability to distinguish between individual samples with probability outputs. Furthermore, this probability learned by AR has the potential to further correct inaccuracies in distances learned by the RQ method.

---

### Author Response · Authors · 2023-11-22
**General Response by Authors, Part 1/2**

Dear reviewers,

Thank you for dedicating time to review our paper and sharing valuable feedback. Here, we succinctly address common queries and clarify misunderstandings raised by reviewer rUYC.

1. We offer a detailed comparison of our approach with previous joint learning methods. Early methods in the literature primarily concentrated on three key aspects: embedding learning to enhance search accuracy, embedding compression (quantization/hashing) to reduce memory cost, and clustering (often involving quantization) or neighborhood graphs to improve search efficiency. However, integrating these essential components individually may lead to suboptimal solutions for large-scale retrieval systems. As a result, there is a growing interest in the study of joint learning.
For example, [1] jointly learns the deep retrieval model (equivalent to an embedding model) and the embedding index (compression). JPQ [2] optimizes encoding and PQ compression jointly using a ranking-oriented loss. SPQ [3] unsupervisedly learns encoding and quantization using a contrastive learning strategy, extending to a semi-supervised manner in [4]. DQN [5] combines PQ and embedding learning but does not optimize the PQ clusters. DPQ [6] incorporates PQ clusters for supervised joint learning, specifically targeting image retrieval. Another DPQ [7] advocates for end-to-end embedding compression. HashGAN [8] and BGAN [9] adversarially learn binary hash codes jointly with the embedding model in an unsupervised way. However, all these prior works focus on end-to-end training of encoding and compression together while overlooking the indexing part (meaning non-exhaustive search), which is challenging to formulate as differentiable. They leverage quantization (compression index) for ANN search (with distance approximation) but still rely on linear scan in their experiments.
In contrast, our approach reframes retrieval as a completely novel end-to-end task, eliminating explicit distinctions between embedding, compression, and indexing. Our method involves offline storage of image identifiers acquired from the proposed tokenizer. Notably, this differs conceptually from quantization, as we do not use the tokenized identifier for distance comparison as in quantization. Retrieving a query's nearest neighbor in our method only involves passing the query through the autoregressive (AR) model. We will incorporate this discussion into the final paper.




[1] Zhang, Han, et al. "Joint learning of deep retrieval model and product quantization based embedding index." Proceedings of the 44th International ACM SIGIR Conference on Research and Development in Information Retrieval. 2021.

[2] Zhan, Jingtao, et al. "Jointly optimizing query encoder and product quantization to improve retrieval performance." Proceedings of the 30th ACM International Conference on Information & Knowledge Management. 2021.

[3] Jang, Young Kyun, and Nam Ik Cho. "Self-supervised product quantization for deep unsupervised image retrieval." Proceedings of the IEEE/CVF International Conference on Computer Vision. 2021.

[4] Jang, Young Kyun, and Nam Ik Cho. "Generalized product quantization network for semi-supervised image retrieval." Proceedings of the IEEE/CVF Conference on Computer Vision and Pattern Recognition. 2020.

[5] Yue, Cao, et al. "Deep quantization network for efficient image retrieval." Proc. 13th AAAI Conf. Artif. Intell. 2016.

[6] Klein, Benjamin, and Lior Wolf. "End-to-end supervised product quantization for image search and retrieval." Proceedings of the IEEE/CVF Conference on Computer Vision and Pattern Recognition. 2019.

[7] Chen, Ting, Lala Li, and Yizhou Sun. "Differentiable product quantization for end-to-end embedding compression." International Conference on Machine Learning. PMLR, 2020.

[8] Dizaji, Kamran Ghasedi, et al. "Unsupervised deep generative adversarial hashing network." Proceedings of the IEEE conference on computer vision and pattern recognition. 2018.

[9] Song, Jingkuan, et al. "Binary generative adversarial networks for image retrieval." Proceedings of the AAAI conference on artificial intelligence. Vol. 32. No. 1. 2018.

---

> ### Author Response · Authors · 2023-11-22
> **General Response by Authors, Part 2/2**
>
> 2. We add a comparison with recent supervised deep quantization methods on ImageNet, including the state-of-the-art methods ADSVQ and DPQ. Notably, these methods exclusively utilize 100 classes from ImageNet, as indicated in their original papers, making it a substantially easier case. To ensure a fair comparison, we reran our model for 100 classes, whereas the results in our paper are based on the full 1000 categories of ImageNet. The table below illustrates the superiority of our approach.
>
> |  Method  |   mAP@1000 for 32 bits (equivalent to the length of our identifier)  |
> | --------:| ------------:|
> |   IRGen     |     93.52     |
> |DPQ [1] | 87.70 |
> |   ADSVQ [2]  |     75.21     |
> |   DTQ [3]   |     68.12     |
> |ADSH [4] | 63.36|
> |HashNet [5]| 58.71|
>
>
> [1] Klein, Benjamin, and Lior Wolf. "End-to-end supervised product quantization for image search and retrieval." Proceedings of the IEEE/CVF Conference on Computer Vision and Pattern Recognition. 2019.
>
> [2] Zhou, Chang, Lai Man Po, and Weifeng Ou. "Angular deep supervised vector quantization for image retrieval." IEEE Transactions on Neural Networks and Learning Systems 33.4 (2020): 1638-1649.
>
> [3] Liu, Bin, et al. "Deep triplet quantization." Proceedings of the 26th ACM international conference on Multimedia. 2018.
>
> [4] Liu, Haomiao, et al. "Deep supervised hashing for fast image retrieval." Proceedings of the IEEE conference on computer vision and pattern recognition. 2016.
>
> [5] Cao, Zhangjie, et al. "Hashnet: Deep learning to hash by continuation." Proceedings of the IEEE international conference on computer vision. 2017.
>
> 3.  In Figure 5 of the paper, we present a throughput comparison on the Inshop dataset. The table below offers a comprehensive assessment of model capacity and inference cost, focusing specifically on the ImageNet dataset. It is crucial to emphasize that a retrieval system's storage considerations extend beyond just the model; it must also store features for all database vectors, constituting a substantial portion of storage requirements, especially in the context of a large database. While compression can alleviate this concern, the performance is severely compromised due to quantization. In terms of inference cost, our CPU inference cost is 0.30s per image, which is not orders of magnitude larger compared to FT-CLIP with linear scan. Although FT-CLIP with SPANN lowers the inference cost, it comes at the expense of a substantial drop in performance. Our GPU inference cost, tested on a Tesla P100 PCIe 16GB GPU, is 0.037s/image, which is near-real-time, considering practical retrieval systems at the order of 10 milliseconds. Moreover, our approach demonstrates competitive performance compared to FT-CLIP with linear scan, and significantly reduces the memory cost. For inference speed, we anticipate that the prevalent reliance on GPUs for generative models will lead to newer GPU generations with increased processing power, more efficient memory management, and improved hardware support, thereby enhancing the inference speed of our autoregressive retrieval. Additionally, our model, being GPT-based, can leverage rapidly developed acceleration techniques designed for GPT.
>
> |  Method | Model (#Params) |  Database storage  |
> | -------:| ----------------- |  --------------:|
> |      IRGen | ViT-base encoder (86M)+ 24-layer decoder (307M)  | 4.9MB： 4 bytes (id)/image with 1281167 samples    |
> | FT-CLIP | ViT-base (86M)    | 3.67GB：  768x4 bytes (feature)/image with 1281167 samples  |
>
>
> |  Method | Search         | Inference cost | mAP@100    |
> | -------:| -------------- | --------------:| --- |
> |      IRGen | Autoregressive |  0.037s/image|   76.0  |
> | FT-CLIP | Linear scan (SPANN)   |   0.14s/image (0.007s/image) |   77.0 (65.5)  |
>
> 4. We itemize our contribution as follows:
> (1) To the best of our knowledge, our work represents the first effort in employing a sequence-to-sequence generative model to address image retrieval in a fully end-to-end manner that directly generates the nearest neighbor's identifier givent the query.
> (2) We introduce a semantic image tokenizer explicitly designed for image retrieval to obtain image identifiers, facilitating autoregressive training. It is essential to note that this tokenizer differs significantly from previous quantization models, as our identifier solely represents the image and is not employed for distance computation, as is the case in quantization.
> (3) Extensive experiments show that our approach exhibits substantial improvements in results, achieving new state-of-the-art results. We have demonstrated its scalability to million-scale datasets and provided throughput number, showcasing its near-real-time performance.

---

### Meta-Review · Area_Chair_C7Eg · 2023-12-05

**Metareview:**

This work proposes to reframe image retrieval as a variant of generative modelling, in line with recent works that have made the same reframing for textual-document retrieval and other modalities. Specifically, the approach learns an encoder-decoder model that takes an image as input and outputs a sequence that serves to identify nearest neighbours. All reviewers found this work marginal wrt novelty and clarity. While the approach was at least somewhat novel, most reviewers felt that important sections of the paper were very unclear, e.g. the beam search formulation, and what data was used for pre-training. The authors did not submit an updated pdf, making it impossible to assess whether important reviewer concerns about clarity and comparisons to related work were adequately addressed. Given this, the AC recommends rejection, and encourages the authors to rework this paper and submit to a future venue.

**Justification For Why Not Higher Score:**

See above

**Justification For Why Not Lower Score:**

n/a

---

### Decision · Program_Chairs · 2024-01-16

Reject